



# A Methodology For Optimal Designing Of Monitoring Sensor Networks For Tsunami Inversion

Joaquín Meza[1], Patricio A. Catalán[1,3,4], and Hiroaki Tsushima[2]

[1]Departmento de Obras Civiles, Universidad Técnica Federico Santa María, Valparaíso, Chile.
[2]Meteorological Research Institute, Japan Meteorological Agency, Tsukuba, Japan
[3]Centro de Investigación para la Gestión Integrada del Riesgo de Desastres (CIGIDEN), Santiago, Chile.
[4]Centro Científico Tecnológico de Valparaíso-CCTVal, Universidad Técnica Federico Santa María, Valparaíso, Chile.

*Correspondence to:* Joaquin Meza (joaquin.meza@usm.cl)

**Abstract.** A methodology to optimize the design of an offshore tsunami network array is presented, allowing placement of sensors to be used in a Early Tsunami Warning System framework. The method improves on previous methods by including multiple tsunami parameters as a measure of the predictive accuracy through a single cost function. The use of different tsunami parameters allows for a network which is less subject to biases found when using a single parameter. The resulting
network performance was tested against an historical event, suggesting that having such a network in place could have provided meaningful information for the hazard assessment. The low number of sensors required may be useful in implementing such networks in places where funding of denser arrays might be of concern.

## 1   Introduction

Over the last decade, several tsunami events, such as those of Maule in 2010 and Tohoku in 2011, have further demonstrated
the catastrophic and widespread potential for death and destruction inherent in tsunami waves and, at the same time, the need to improve the reliability of Tsunami Warning Systems (TWS). The 2010 Maule earthquake ($M_w = 8.8$) generated a tsunami that caused severe damage and loss of life in coastal communities (e.g. Fritz et al., 2011), and highlighted the consequences of an ineffective alert (Soulé, 2014). This event, and the improvements observed in the cases of other events, has reaffirmed the importance of improving timely evacuation warnings, which ar considered to be one of the most effective ways to reduce the
loss of human lives and damage to coastal communities (Okal, 2015).

Improving a TWS aims to the overarching goal of improving delivery of a timely and meaningful evaluation of the hazard to authorities, and through them to the population at large, with the main objective to trigger evacuation. While the role of education is usually considered the cornerstone for successful responses, the role of accurate information regarding the actual
hazard is also relevant (Okal, 2015; Bernard and Titov, 2015). Over the years, improving the hazard assessment has followed different approaches. One common key aspect is maximizing the lead time of the warning relative to tsunami arrival to the area of interest.



For far field tsunami forecasting, where coastal tsunami impact can be evaluated well in advance of arrival time, the approach is to combine dense monitoring of the actual tsunami along its propagation path, and take advantage of this information to estimate the hazard. For instance, the United States has a worldwide network of offshore tsunami observatories, which are located near several subduction zones at distances equivalent to 30-60 minutes of tsunami wave travel time from expected tsunamigenic earthquake sources. This network was built by the Pacific Marine Environmental Laboratory of the National Oceanic and Atmospheric Administration, and it is termed the Deep-ocean Assessment and Reporting of Tsunamis (DART) system (e.g. Spillane et al., 2008; Percival et al., 2011). Ocean Bottom Pressure Sensors (OBPSs) record the tsunami and data are transmitted to data centers in real time by means of communication satellites (González et al., 2005). Data are then used as input to invert an estimate of a tsunami source (Percival et al., 2011), which allows forecasting a set of tsunami hazard products (Bernard and Titov, 2015).

For near field tsunamis forecasting, on the other hand, the short time interval between the tsunami generation and arrival may be insufficient to perform tsunami source evaluations and propagating forward the tsunami. As shown by Williamson and Newman (2018), in some places first arrivals can occur in as little as 5 min. For example, Aránguiz et al. (2016) anecdotally reported arrivals in less than 8 min for the $M_w 8.4$ Illapel earthquake and tsunami. A relevant aspect of the problem is the finite time required to acquire data sufficient to allow for an inversion of the source. However, owing to the significantly different propagation speeds between ocean surface and seismic waves, the preferred approach is to rely on seismic data alone. Although some seismic source solutions can be obtained in very short time, the inherent uncertainties can affect the accuracy of tsunami estimates which may limit its applicability (Cienfuegos et al., 2018). Moreover the high computational cost of computing tsunami propagation and inundation, at least to date in operational systems, has prompted the use of table look-up procedures on data sets of scenarios calculated a priori stored in a database (e.g. Gica et al., 2008; Kamigaichi et al., 2009). This approach is used in countries such as Japan, Australia, and Indonesia, and recently, in Chile.

Nevertheless, it has been noted that including tsunami data in the inversion process usually leads to improved results in estimating the tsunami hazard (e.g. Gusman and Tanioka, 2014; Gusman et al., 2014; Cienfuegos et al., 2018). Hence, it highly desirable to incorporate tsunami data as early as possible in a TWS framework. These observations can be acquired from tide gage data (Satake, 1987; Satake and Kanamori, 1991), satellite altimetry (Arcas and Titov, 2006; Hirata et al., 2006), deep-ocean tsunameters (Titov, 2009; Percival et al., 2011) and cabled ocean bottom pressure sensors (OBPSs, Baba et al. (2004); Tsushima et al. (2009, 2012)). Among these, the use of coastal stations (tide gages) is typically not considered for early warning, owing to the nil lead time and possible influence of coastal bathymetry in the hydrodynamics, hence obfuscating the effect of the tsunami source. Hence, offshore tsunami data from tsunameters or OBPSs seem to be the most appropriate for TWS.

Williamson and Newman (2018) analyzed the possible coastal locations where sensors can provide a meaningful lead time, in a near field setting. These locations concentrate along narrow bands that run roughly parallel to subduction zones. In addition, one of the most important factors for tsunami forecasting based on offshore tsunami data is the array configuration of



tsunami stations. The accuracy of tsunami source reconstruction strongly depends on the azimuthal coverage of recording stations with respect to the source area (e.g. Pires and Miranda, 2001; Piatanesi et al., 2001; Bernard et al., 2001). In other words, accuracy depends whether the sensors are located close to the main beam of tsunami energy. In addition, when an adequate amount of sensors surround the tsunami source area with substantial azimuth coverage, it is possible to determine with greater

detail the tsunami source. Hence, a dense network of sensors might be required. To date, only four developed countries, the U.S., Japan, Canada and Oman, are able to mantain dense cabled sensor networks. A major handicap is the high cost of the sensors themselves and their installation, as well as their subsequent operation and maintenance. In the case of Japan, they have a few submarine cabled seafloor observatory networks (e.g. S-net systems, DONET1 and DONET2) that provide data in real time. A 1000 km cabled observatory with 164 bottom pressure sensors was installed at an estimated cost of US\$ 500M

(Bernard and Titov, 2015). This high cost may pose a significant hurdle for developing countries along subduction zones, such as Chile or other countries on the eastern Pacific seaboard, where the seismogenic zone is extensive. However, despite this potential economic hurdle, it is relevant to study whether less dense arrays could provide a working solution at lower cost.

The placement of tsunameters has been based on expert judgment which considered technical aspects such as variability and
location of seismic sources (Schindelé et al., 2008), travel time (Schindelé et al., 2008; Omira et al., 2009; Williamson and Newman, 2018), financial (Araki et al., 2008), and legal aspects (Abe and Imamura, 2013), among others. Bernard et al. (2001) suggested that along-strike sensor spacing of about 175-700 km is required to characterize the main energy beam of $M_w \sim 8.0$ events with just three sensors. The actual spacing will depend on the allowed elapsed arrival time to the sensors. In order to improve the tsunami warning efficacy to design tsunamis observation networks, Schindelé et al. (2008); Omira et al. (2009)
took into account the seismicity to identify possible sources, and design optimal placement of sensors using as target function the travel time and a set of delays. In particular, Schindelé et al. (2008) use 16, evenly spaced sources spanning a long stretch of coast of the Western Mediterranean basin, which are used to test the efficacy of a set of two predefined arrays. 13 tsunameters spaced about 50-90 km yield the best results. Omira et al. (2009) use just a single scenario for each of five tsunamigenic zones, but on a smaller domain, resulting in a array consisting of just three sensors. On the other hand, Spillane et al. (2008) carried
out an optimization approach to place DART buoys in Aleutians and Alaska regions. Time detection was the main restriction to place tsunameters, where it was found that adding more than three sensors did not improve the results significantly. Mulia et al. (2017a) used a similar approach, but a dimensionality reduction approach was incorporated to initiate the optimization. Unlike the previous cases where the goal is to address a wide range os sources, Mulia et al. (2017a) and Mulia et al. (2017b) aim at the best placing of sensors to characterize a large–magnitude, target scenario. Here, the focus is to capture with great
detail the characteristics of non-uniform slip by maximizing the accuracy of inverting a set of stochastic scenarios on a predefined domain. Inverted slips are compared against each stochastic source, under the assumption that tsunami parameters will be well determined if the earthquake source is well retrieved. Hence, no evaluation of the tsunamis are performed. An initial set of 30 sensors is obtained, which can be reduced to 23 sensors at specific locations after optimization. This highlights that in pursuing the detailed structure of slip, a large number of sensors is required. Finally, Saunders and Haase (2018) proposed an
improvement in slip characterization for the Cascadia Subduction Zone testing five different, dense sensor arrays. To assess the



quality of results, the root mean square, maximum fault slip, tsunami amplitude at the coast and percentage of coastline hit by the high amplitudes waves between recovered from inversion and input data were compared, which expands on the preferred use of travel times as assessment parameter.

In the perspective of cost-performance ratio, it might be of interest to establish the minimal number and location of sensors required to provide a reasonable estimation over. This purpose cannot be reached without taking into consideration the azimuthal coverage and the time required for observation, which strongly depends on the spatial relationship between the tsunami source and the tsunameters (e.g. Bernard et al., 2001). Although it is true that for both tsunami warning and scientific purposes, more data is always desirable, the number of deployments can limited for economic and technical reasons. For de-
veloping countries, a solution balancing cost and precision might be more appealing.

    Consequently, in the present work, a methodology for estimating the optimal placement of a small numbers of sensors is used and tested. Tho carry out an objective comparison, three different tsunami parameters are considered in unison to find the optimal configuration. The performance of the resulting network is evaluated by simulating its application to the tsunami of
Pisagua 2014 (Catalán et al., 2015).

## 2    Methodology

The overall objective is to determine an optimal array configuration of offshore tsunami sensors for near-field tsunami forecasting, to be used as a data source for a tsunami inversion technique. In what follows, the analysis will focus in optimizing
the forecast accuracy of relevant tsunami parameters, and leaving aside other possible considerations such as cost or technical constraints on sensor placement. The methodology assumes the sensor are capable of providing free surface data at sufficient temporal resolution, therefore it is considered independent of the type of sensor. The methodology is similar in scope to previous studies (e.g. Schindelé et al., 2008; Omira et al., 2009), but here other tsunami parameters are included in the assessment.

The method builds on the premise that, in order to determine the tsunami source, an inversion procedure must be implemented. In this case, the "tsunami Forecasting based on Inversion for initial sea-Surface Height" (tFISH) (Tsushima et al., 2009) is used. To carry out the inversion analysis, offshore tsunami waveform data are inverted (in real time in an operational setting) to estimate the initial distribution of the sea surface displacement. Coastal tsunami waveforms are then synthesized by linearly combining pre-computed tsunami Green's functions weighted by the resulting initial distribution. The method allows
for considering the coseismic displacement in the inversion procedure, thereby allowing to place sensors in the seismogenic zone. More details about tFISH can be seen in Tsushima et al. (2009, 2012).





The use of Green's functions is preferred over directly modeling individual events to target sensor locations (e.g. Schindelé et al., 2008) because it allows for testing and comparing a large number of sensor configurations at low computational cost, provided a quantitative parameter or cost function is defined for the comparison. To compute the Green's Function for each subfault patch, a numerical scheme of the linear long-wave equations is used to propagate each tsunami elemental source to a

set of predefined forecast points. The initial sea surface model for each Green's function is represented by a Gaussian function. Overlapping Green's functions are considered to express smooth variations of sea-surface displacement with a finite number of discrete elements (Aida, 1972). For the purposes of the present work, a set of nearly 1000 Greeen's functions is considered. Each of them is of dimensions of **700 x 700 arc seconds**, with its centers spaced **0.15 arc-deg**. They cover an area spanning about $7° \times 3°$ (latitude and longitude), consistent with the so called Northern Chile Gap (Comte and Pardo, 1991; Metois et al.,

2013). In what follows, although an inversion process can benefit by other data sources such as Deep ocean Assessment and Reporting of Tsunamis (DART) buoys, these are not considered in the analysis under the premise that the area of interest is developing a completely new system. On the other hand, for the particular case of Chile, the location of the existing DART buoys in the area of interest is such that requires longer observation times than the ones studied here (Williamson and Newman, 2018).

Tsunami free surface time series, $\eta(t)$, are computed at a set of offshore and coastal observation points for a prescribed duration of the event, $T$. The former are possible locations of a single observing station or sensor, and a subset of sensors is termed a sensor array. The latter correspond to tide gages, which are used to evaluate the predictive performance at the coast. Hence, the overall procedure considers that given a source event, and for a given observation array, it is possible to estimate an inverted tsunami source solution. This solution is then propagated to the coastal points, where a set of relevant parameters are

assessed to evaluate the quality of the solution by comparing it against a target scenario calculated independently.

To this end, the error in three tsunami parameters is estimated. As others (e.g. Schindelé et al., 2008; Omira et al., 2009), tsunami arrival time is considered a key parameter, as it is essential in a TWS framework to provide a timely hazard assessment. However, the definition of arrival time is relatively loose and could refer to different stages of the tsunami, such as the first

exceedance of a threshold, the first local crest, first initial trough of N-waves, and others (e.g. Hayashi et al., 2011). Here, two different definitions are considered in order to make the analysis more robust. The first definition defines the time $T_1$ at which the tsunami exceeds a certain threshold

$$T_1(\eta) = \min(t \in (0, T) \mid \eta(t) > \epsilon). \tag{1}$$

Although this definition is not used in operational TWS, it is considered a relevant estimate for the time required to trigger

a warning status. However, considering that in some cases, the tsunami time series might not exceed the threshold, a second arrival time is defined as the time when a proxy for the slope of the free surface exceeds a certain threshold

$$T_2(\eta) = \min\left(t \in (0, T) \mid \frac{\partial \eta}{\partial t} > \delta\right). \tag{2}$$


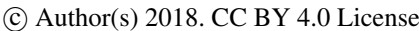


In this case, the idea is to establish a measure of the rate of change of the tsunami signal as an early proxy for the first local maximum. In doing this, it is assumed that the actual free surface slope is proportional to the rate of change

$$\frac{\partial \eta}{\partial x} \propto \frac{\partial \eta}{\partial t}. \tag{3}$$

However, the accuracy in predicting the free surface time series is also relevant for a TWS. A relevant parameter is the

5 maximum tsunami amplitude (here denoted as $H$), which is the parameter used to categorize the hazard in existing TWS. The maximum tsunami amplitude is estimated as

$$H = \max(\eta(t) - \eta(0) \mid t \in (0, T)). \tag{4}$$

It is of note that neither arrival time nor the maximum tsunami amplitude consider the accuracy in retrieving the shape of the waveform. Hence, the skill estimator is also introduced ($S_k$), as this index is commonly used to evaluate numerical perfor-

10 mance of a model (e.g. Hampson et al., 2011)

$$S_k = \sqrt{\frac{\frac{1}{n}\sum_{i=1}^{n}\left(\eta_{\text{for}} - \eta_{obs}\right)^2}{\frac{1}{n}\sum_{i=1}^{n}\left(\eta_{obs}\right)^2}}, \tag{5}$$

where $\eta_{\text{for}}$ and $\eta_{obs}$ are the time series forecasted from the inverted source, and the target or observed source. $n$ represents the number of time steps of the time series.

To provide a common comparison basis for all possible sensor configurations, each parameter is normalized to the range 0-1, and a saturation value equal to one is defined, as follows

$$\Delta T_m = \min\left\{\left|\frac{T_m(\eta_{obs}) - T_m(\eta_{for})}{T_m(\eta_{obs})}\right|; 1\right\}, \tag{6}$$

$$\Delta H = \min\left\{\left|\frac{H(\eta_{obs}) - H(\eta_{for})}{H(\eta_{obs}) - \eta(0)}\right|; 1\right\}, \tag{7}$$

$$S_k = \min\left\{S_k; 1\right\}. \tag{8}$$





where $m = 1, 2$ applies for the different time travel parameters. The saturation value indicates errors $\Delta T_m$ or $\Delta H$ exceeding 100% are not penalized in excess. In the case of the skill, $S_k = 1$ indicates that the magnitude of the error is comparable or greater than the observed values. Zero values mean a perfect fit on the indicator. In addition, whether each quantity is under or overestimated is not considered relevant and absolute values are used.

To combine all estimators into a single quantitative estimate, a forecasting accuracy function is introduced

$$F_{i,j}(\Delta T, \Delta H, S_k) = \alpha \Delta T_{i,j} + \beta \Delta H_{i,j} + \gamma S_k i, j, \tag{9}$$

which allows quantifying the total error of the estimation at the forecasting point $j$ given an offshore array $i$. $\alpha, \beta$ and $\gamma$ are weights that allow for user-defined tuning of the relative importance of each parameter. As a first estimation, $\alpha = 0.40, \beta = 0.40$ and $\gamma = 0.20$ are considered, owing to the larger weight attributed to arrival time in previous research. Moreover, as to couple both definitions of arrival times, the error estimator associated to this parameter was considered as the average of each percentage error, $\Delta T = 0.5(\Delta T_1 + \Delta T_2)$. In this way, it is possible to quantitatively compare the performance of all sensor arrays at any given forecast point.

However, it is of interest to test global accuracy of the method at several forecast points simultaneously. The overall performance is computed by simply adding up the individual results at forecast points, given a sensor array $i$

$$EG_i = \sum_{j=1}^{N} F_{i,j}(\Delta T, \Delta H, S_k), \tag{10}$$

where $N$ is the number of forecast points. Upon calculating the global error by means of Eq. 10 considering all possible offshore arrays, the candidate array is selected as that featuring the minimal global error ($\min(GE_i)$). However, it is possible for Eq. 10 to be biased by a few forecast points. To this end, it is imposed that the chosen array must consider that the error of each forecasting point $j$ must be less than a certain threshold ($F(\Delta T, \Delta H, S_k)_{i,j} < \mu, \quad \forall j$). This is equivalent to ensuring a minimal forecast capacity for each forecasting point $j$.

Regarding the construction of a sensor array, prior research indicates that two to four observation stations are capable of constrain the source parameters quite well if the stations are optimally located relative to the main tsunami energy beam, and adding more data does not significantly improve the inversion results (e.g. Percival et al., 2011; An, 2015). On the other hand, the relative distance of the sensors will depend on the time allowed to record the tsunami (e.g. Bernard et al., 2001). Hence, given an earthquake, it is possible to define the area (thus time, if the tsunami propagation speed is known) over which the tsunami waves have already propagated away from the source (e.g. Williamson and Newman, 2018). A minimum of two sensors must be considered inside this area, henceforth termed listening area $A_l$. Conversely, the listening area can be related to the data observation time, $T_0$, representing the time the TWS allows for recording of the tsunami before performing an inversion.





To test the performance of the sensor arrays, a set of tsunami sources must be considered. While it is possible to test a large set of sources over a large domain as in Schindelé et al. (2008), here the focus is set on using sources located at the extreme of the area of interest, under the assumption that these will correspond to the worst scenario for the arrays owing to the decrease in sensitivity of detection as sensors are located further away of the main tsunami energy beam. Finally, once a sensor array

has been defined, the candidate network is tested using other locations for synthetics sources and past events.

## 3   Data and Sensor Array Design

For the purpose of testing the possible sensor arrays, two tsunamigenic earthquakes, with seismic moments ($M_w$) 8.3 and 8.5 respectively, are considered as scenarios. These were determined by Cienfuegos et al. (2014) as representative events for the Northern Chile Gap. These are located just in front of Arica, near the Chile-Perú border, and just north of Peninsula Mejillones,

as seen in Fig. 1. They also flank the rupture area of the 2014 Pisagua Earthquake. These scenarios were estimated from the interseismic coupling model of Chlieh et al. (2011), information on interseismic slip rates and convergence. The choice of the scenarios is based on the assumption that these constitute the worst case for performing an inversion. Hence, the optimal array should be capable of detecting each scenario and others located in between them.

A set of $\sim 1000$ unit tsunami sources were distributed uniformly over an area spanning $160 \times 680$ km. Tsunamis were propagated using the tsunami model JAGURS (Baba et al., 2016) to eight coastal stations (coinciding with existing tide gages operated by the National Hydrographic and Oceanic Service of the Chilean Navy, SHOA, by its acronym in Spanish), and also to an array of virtual observation points, each of which denote the possible location of a possible sensor. Bathymetry considered GEBCO global data with spatial resolution of 30 arc seconds, since this resolution is sufficient if the subfault size is larger than

$40 \times 40$ [km] (An, 2015). All Green's functions were calculated in advance and stored in a database, from which appropriate functions were extracted and used for the inversion and forward calculation.

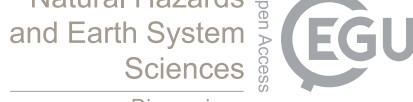



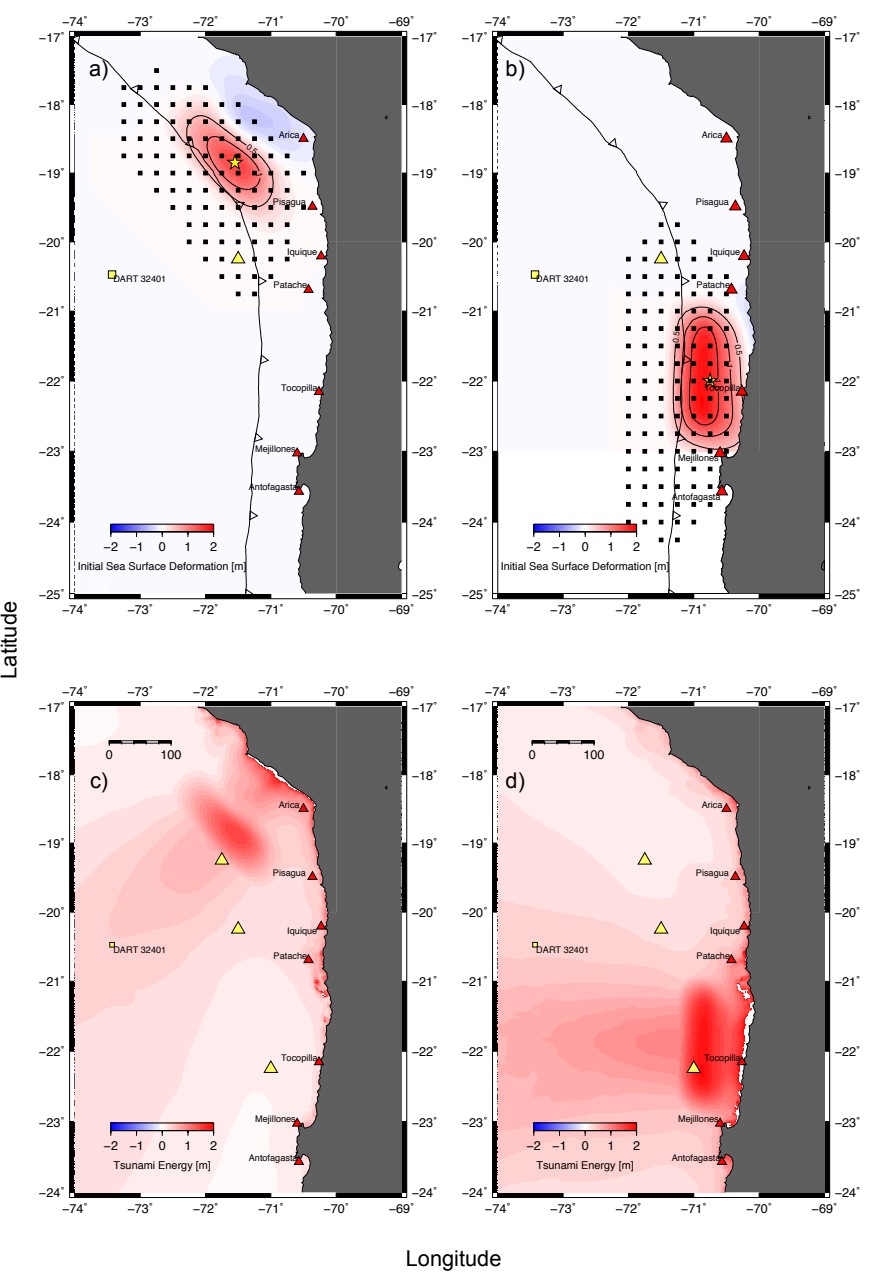

**Figure 1.** (a-b). Sea surface deformation for the scenarios considered. Stars indicate epicenters. Black squares show virtual observation points offshore, red triangles indicate the forecasting points, and the yellow triangle represents the fixed sensor. (a) shows a $M_w$ 8.3 and (b) a $M_w$ 8.5 scenarios, located in the northern and southern end of the the Northern Chile Gap, respectively.(c-d) are the corresponding maps of tsunami energy. In (c-d), the final sensor configuration is shown.





The relative position of each sensor to the tsunami scenarios, determine the tsunami arrival time to the sensor but also the elapsed observation time of data usable for inversion. In existing inversion procedures, each sensor uses a different observation time, sufficient to gather at least a quarter of the initial tsunami waveform. Here, the observation time is defined as to determine an area such as any sensor located inside of this area could record at least half a tsunami wavelength ($0 < t < T_0$). This

defines the listening area, $A_l$. It is noted that the observation time is also restricted by the tsunami arrival to the coast, in order to provide sufficient lead time for an eventual warning. As a starting point, $T_0$=10 min is used, consistent with the observed tsunami arrival for 2014 Pisagua tsunami (Catalán et al., 2015).

Each of the tsunami scenarios is propagated to determine its propagation time and estimate the listening area. Considering

the time restriction imposed, there is relatively small region where the listening area $A_l$ of both scenarios overlap. Hence, it is possible to place a sensor that would serve both the northern and southern sections, located in the outer rise offshore of Iquique (see yellow triangle in Fig. 1). To find the location of the other sensors, the listening area for each scenario is discretized regularly every 0.25 arc degrees ($\sim$ 30 km) in both latitude and longitude, which is significantly less than the tsunami wavelength in the area. Each node is considered a possible sensor location. In addition, the nodes are to be located at depths large enough

for tsunami nonlinear effects to be considered negligible, thereby ensuring consistency with the assumemption by the inversion algorithm. As a consequence, 99 and 113 possible nodes are defined in the northern and southern parts, respectively, denoted as black squares in Fig. 1. The difference in the number of nodes is due to the differences in the source dimensions, and in tsunami celerity arising from the bathymetry, which in unison determine different listening areas. A sensor array was defined by the pairing of the common sensor and each of the possible nodes.

To carry out the analysis, each scenario is propagated forward towards the nodes, and also to coastal forecast points (FPs), using the model COMCOT. The time series at nodes are considered as target (observed) time series, whereas the use of coastal forecast points is aimed at establishing the quality of the assessment. The use of a different tsunami model in propagating the signal and in preparing the database of Green's functions allows for differences in the target tsunami time series to those of

the database, which is considered sufficient to replicate some of the inherent tsunami variability. No other source of variability, such as noise, is considered. It is also assumed that tsunami-tide interactions do not play a significant role in this area.

For each of the 212 sensor arrays (99 and 113, for each scenario), the observed tsunami time series are used to invert the tsunami source. Once the source is determined, linear combinations of the Green's functions consistent with the source weights

are estimated in the coastal points, and the observed and estimated time series are compared, using the parameters defined in the previous sections, for example, Eq. 9.





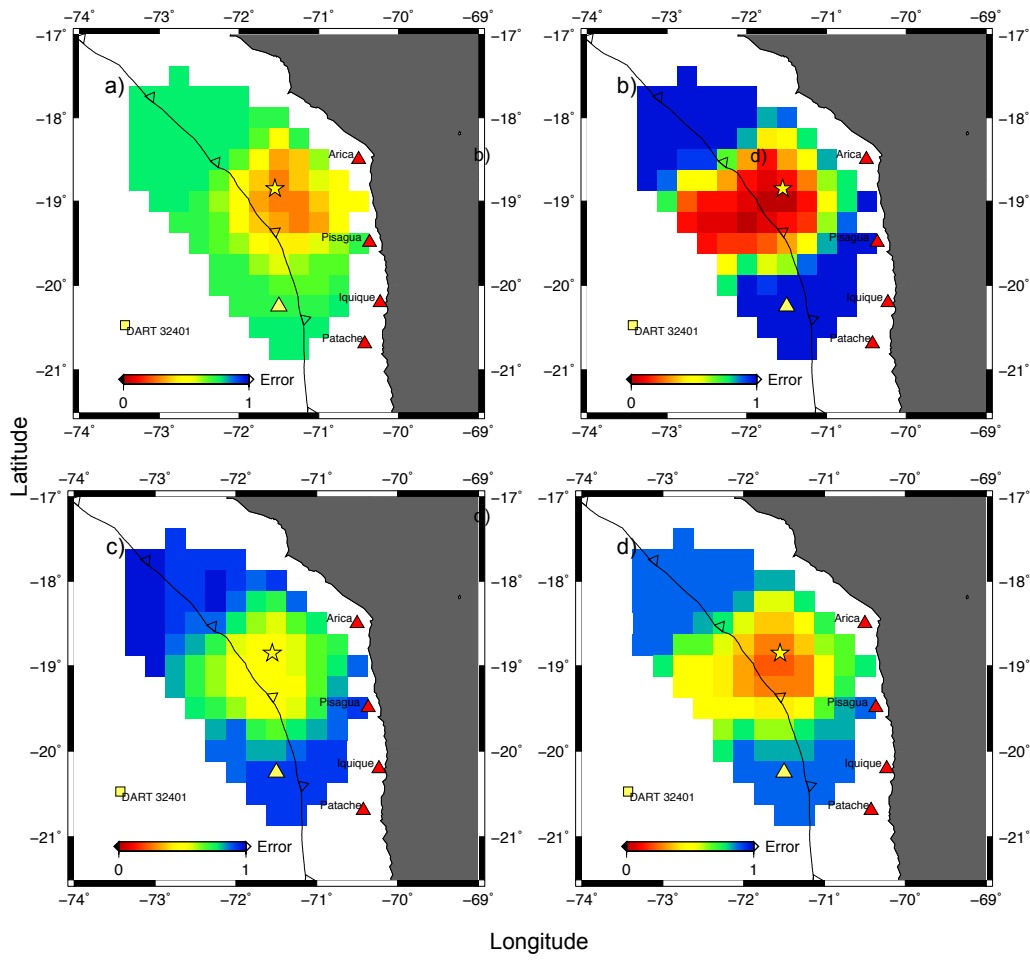

**Figure 2.** Error distribution for sensor arrays in the northern section. Star indicates the scenario epicenter, the thin black line is the location of the trench, the yellow triangle denotes the location of the fixed sensor, and each grid point in the color map corresponds to a movable sensor. (a) $\Delta T$ (Eq. 6); (b) $\Delta H$ (Eq. 7); (c) $S_k$ (Eq. 8); and (d) Total error $F_{i,j}$ (Eq. 9) estimated at Arica.

## 4 Results

Coastal points are located in front of the main cities where a tide gage is present, at 200 meters depth, consistent with the linear propagation used by tFISH. Waveforms obtained through inversion were compared with observed waveforms by means of the Eq. 9.





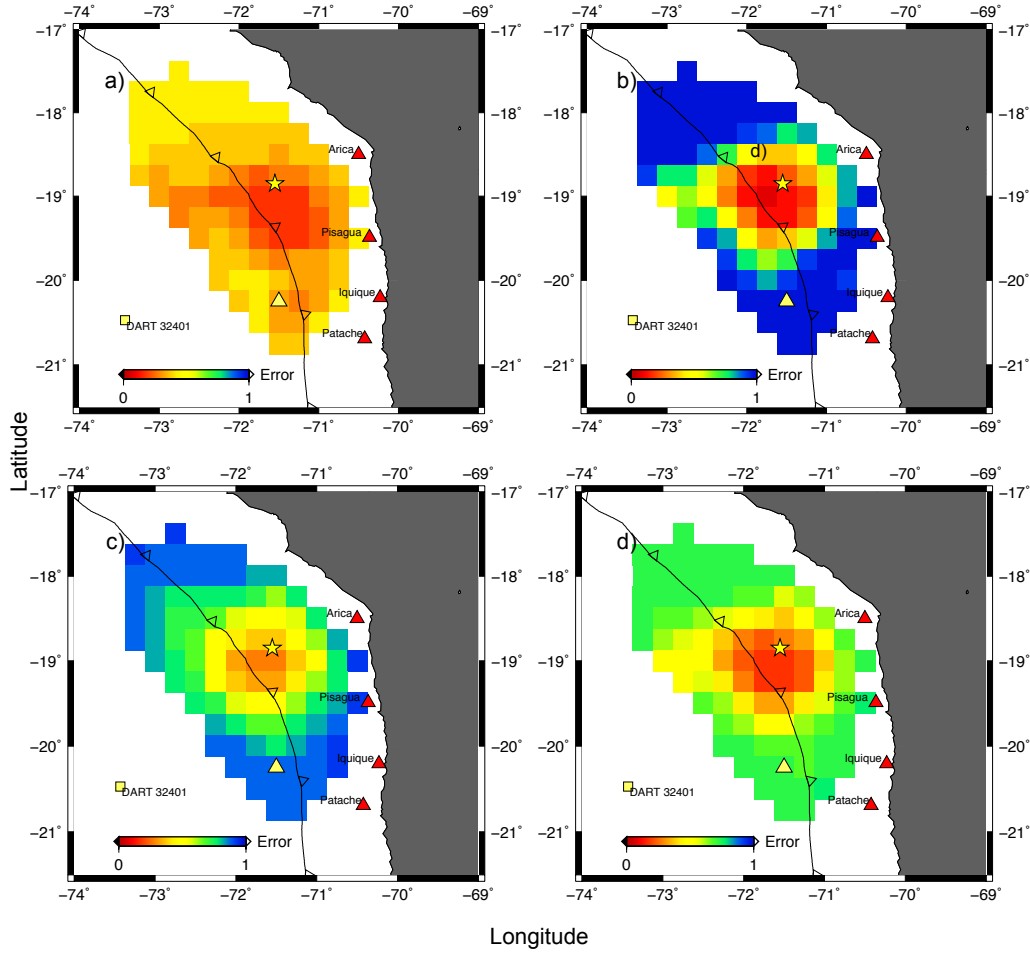

**Figure 3.** Error distribution estimated at Patache. Same key as Fig. 2.

In Figs. 2 and 3, the accuracy of the estimation is presented as a space map of the value of each of the error estimates defined by Eqs.6 - 9, , presented as a colorscale. Each grid point corresponds to the location of the second sensor of the array. Smaller values (red colors) indicate better accuracy. Results are presented when the evaluation is carried out using as a coastal forecast point $j$ at Arica and Patache, respectively.

The errors in arrival time, $\Delta T$, (Fig. 2a and 3a), show better results when the sensors are close to the forecast point. This could be due to a better correspondence in arrival time between offshore and coastal waters. Arica and Patache show a distinct behavior, where Arica is more sensitive to the sensor location, with an error of $\sim 35\%$ on average, and accuracy reaching





saturation at some locations ( errors equal or greater than 100%). Patache is less sensitive and shows a better accuracy over-all, partly because it benefits for the fixed sensor being located in front of it, making the results less sensitive to the other sensor.

The error in amplitude, $\Delta H$ is shown in Fig. 2b and 3b). Unlike the case of the arrival time, the performance at Arica and
Patache are similar, suggesting that the movable sensor is more relevant in determining the tsunami amplitude, allowing for a better estimation. This pattern is related to the directivity of the tsunami energy radiation. When the tsunami source has an aspect ratio (length to width) greater than one, most of the tsunami energy is radiated perpendicular to the major axis of the source (Kajiura, 1970, 1972). Consequently, errors smaller than 17% are found in some cases where the sensor is located close to the main energy beam. In contrast, when the sensor is located parallel to the major axis, where weaker amplitudes are
radiated, tsunami wavefronts used as input do not contain information about the maximum tsunami energy so that the initial sea surface is underestimated and leads also to underestimation of the amplitude at FPs. When sensors are located outside the main energy beam, accuracy decreases rapidly down to the saturation limit. The location of the fixed sensor does not allow for improved inversions, explaining the similar performance at both sites. The difference in performance of the amplitude error and the travel time highlights the need of including the amplitude as a relevant tsunami parameter.

Similar conclusions are obtained from the skill indicator, where higher forecasting accuracy is obtained for sensor arrays with at least one sensor located in the main energy beam. The combination of these individual errors in the aggregate error $_{i,j}$ follows the same trend (Panels d, in Fig. 2-3). The minimum global error is found in the area opposite to Arica, and is influenced by the amplitude error but aggregates the structure of the error in arrival time. On the other hand, in Patache, the
global error distribution shows less contrast between locations than in Arica. These results reinforce the idea that a single error estimate such as arrival time is not sufficient to identify the best placement, but it also shows that using a single forecast point as a reference can be affected by local dependencies. It is important to note there is a smooth transition from bad quality results (blue colors) to good quality results (red colors) for all estimators, which means that the spatial discretization used suffices to capture the error dependencies.

The aggregate of the forecasts at coastal points is estimated using Eq. 10, considering four coastal points for each scenario ($N$=4 in Eq. 10), namely Arica, Pisagua, Iquique and Patache, for the northern scenarios; and Patache, Tocopilla, Mejillones and Antofagasta for the southern scenario. The choice is due to their proximity to the source in each event and their importance in the north of Chile. Fig. 4 shows the global error for both events. As before, the spatial maps show a slight concentration of
improved accuracy near the epicenter of earthquakes, but the variability is reduced owing to the uniforming effect of considering several forecast coastal points in unison in the evaluation. Despite this, it is still possible to identify locations where a sensor could be deployed that yield the best performance. Therefore, the final array configuration is determined by this minimum global error and that the total error for each coastal forecast point, $F_{i,j}$ is less than $\mu = 0.55$, to ensure a good quality at each FP. The selected coordinates of the sensor array are shown in Table 1 for reference.






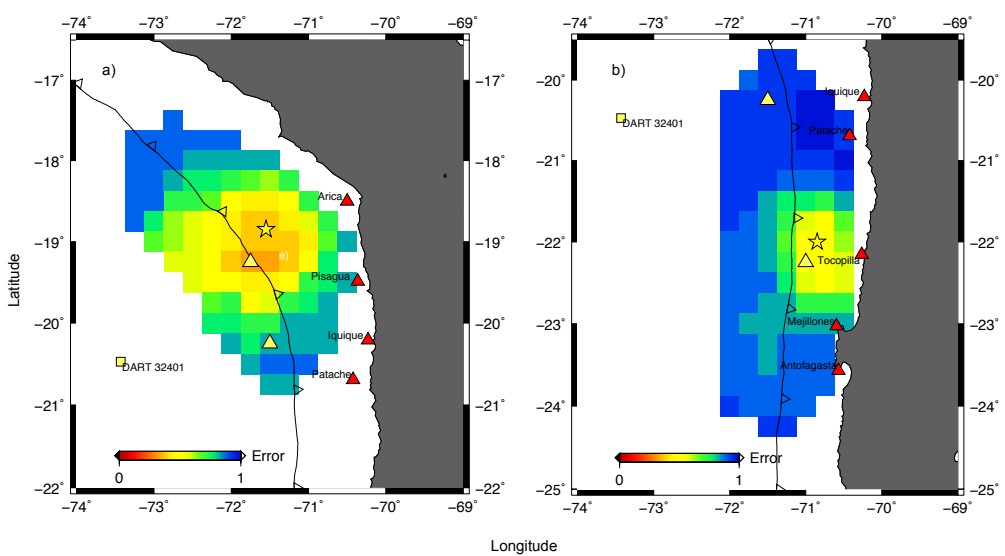

**Figure 4.** Space maps of global error, $EG_i$. Star indicates scenario epicenters, and yellow triangles show selected sensor array. (a) $EG_i$ for the northern scenario, estimated using Arica, Pisagua, Iquique and Patache. (b) $EG_i$ for the southern scenario, estimated using Patache, Tocopilla, Mejillones and Antofagasta.

**Table 1.** Sensor array configuration proposed.

| Name | Latitude | Longitude |
|------|----------|-----------|
| North | 19.25° S | 71.75° W |
| Fixed | 20.25° S | 71.50° W |
| South | 22.25° S | 71.00° W |

## 5  Discussion

The methodology presented allows for an objective comparison of several array configurations, by using a set of relevant tsunami parameters. On the other hand, the method was tested by defining a minimum of two sensors to form a sensor array, under the premise that this represents the least expensive implementation. Testing for arrays comprising a larger number of
sensors, and/or tsunami parameters, is straightforward and are not considered further in this work.





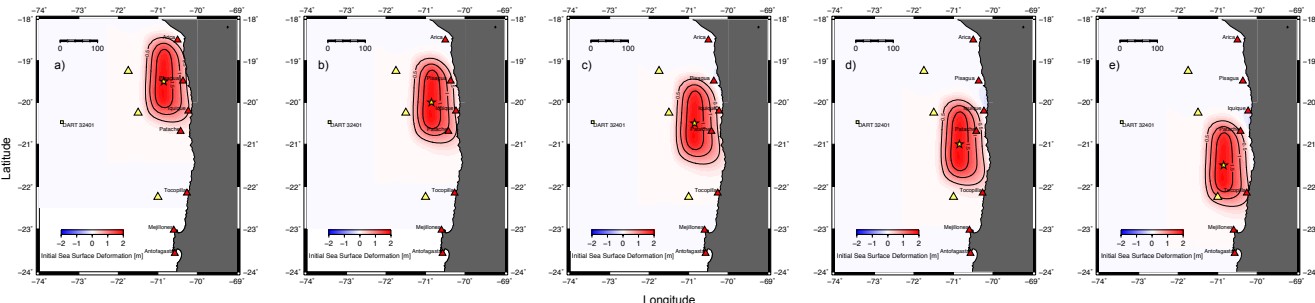

**Figure 5.** Initial sea-surface displacements used in evaluating the chosen sensor array. Red an yellow triangles indicate the forecasting points and sensors respectively. Epicenter are located at latitude (a) -19.5°, (b) -20.0°, (c)-20.5°, (d)-21.0 °; (e)-21.5°

The methodology proposed considered two characteristic scenarios located at the extremes of the area of interest. To evaluate the capabilities of the proposed network in other cases, the southern event ($M_w$ 8.5) was modeled in different locations along the subduction zone, every 0.5 arc degree along strike (see Fig 5). In addition, the observation time was changed (10, 15 and 20 min) to investigate the effect of record length on the forecast.

Fig. 6 shows the matrices of $F_{i,j}$ at each forecast point $j$ as a function of the epicenter location, considering the final array configuration proposed, for different observation times. Regarding the latter, the aggregate error decreases as the observation time increase, as expected, where aggregate errors can reach values as low as 15% when the observation time is 20 min. In the case of Chile, where the seismic zone is located very close to the coast, this may prevent the use of this information as a

trigger for early warning, but it could be considered in the later stages of the emergency cycle. For instance, this information can be used to refine initial hazard assessments derived by other means, such as the existing database of precomputed scenarios.

Regarding the sensor array performance for different scenarios, it is possible a decrease in performance as the scenario is closer to the forecast point. This can be seen as a poor accuracy in the northern FPs (Arica to Iquique) for events with epicenters

at latitudes -19.5° and -20° (See darker blue data in the upper left corner of Fig. 6a), and a worsening of the accuracy as the epicenters are located further south (see the evolution of the error for Tocopilla). This is due to the forecast points being placed close or inside the zone of the predicted coseismic deformation for the actual event, which prompts an early arrival in the observed signal. However, owing to the short listening time, the tsunami source solution results in a source of small spatial dimensions (see for example in Fig. 7b, although for a different scenario). As result, the arrival time yields a large error. In the

other cases, the error averages 40% which is considered a good performance for this observation time. As the observation time is increased, the error in the prediction drops significantly and values as low as 20% error can be obtained.





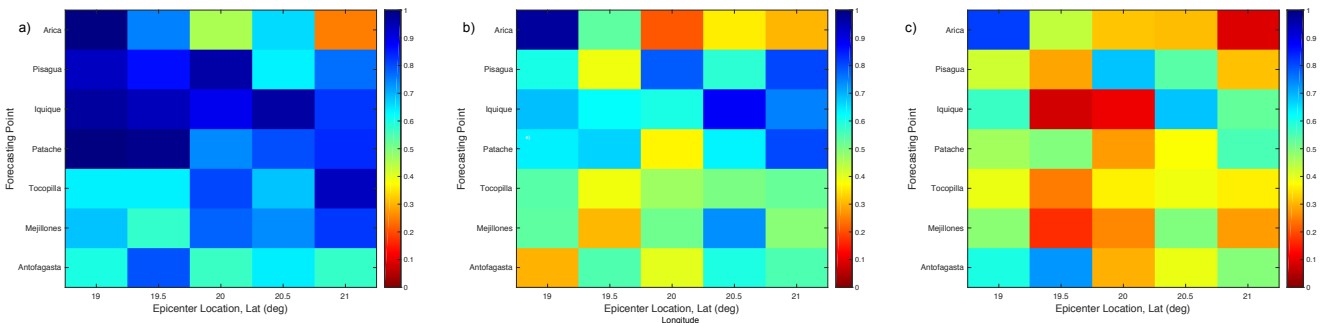

**Figure 6.** Global error $F_{i,j}$ as a function of forecast point $j$ for different sources and data observation times. (a) 10 min, (b) 15 min and (c) 20 min.

As an additional test for evaluating the potential performance of the sensor array, the tsunami of Pisagua on April 1st, 2014 ($M_W 8.1$) was used as source. Synthetic tsunami waveforms at the location of the sensor array were obtained by a numerical simulation of the tsunami using the initial rupture model proposed by Hayes et al. (2014). As before, the accuracy of the assessment is evaluated using the coastal points and different observation times, although in this case, the actual tide-gage records are considered. These were obtained from the IOC Sea Level Monitoring Facility website.

Results are summarized in Table 2 and the initial surface solutions are shown in Fig. 7. It can be seen that within 10 min, the performance of the method is inadequate, with large errors and providing an inverted initial surface that only detects a localized source. As the observation time is increased, the errors decrease significantly, with the reconstructed sea surface condition now having an appropriate extension but a smaller peak displacement. The solution is improved with 20 min of observation time, with errors as low as 8% (40% on average), and the solution including traces of the secondary peak on the north west of the rupture. However, as mentioned before, such a sensor array would have only provided a timely assessment for locations outside the main rupture zone, since the observed tsunami arrival at the tide gages in Iquique and Pisagua was less than 15 min (An et al., 2014; Catalán et al., 2015).

According to these analysis, the proposed sensor array would suffice to provide an adequate assessment of the initial tsunami sea surface. The methodology expands on previous research by incorporating quantitatively different tsunami parameters in a cost function which can be minimized. However, the selection of the weights has ( $\alpha, \beta$ and $\gamma$ ) in the cost function $F_{i,j}$, Eq. 9, was arbitrary. To test the influence of these weights in the assessment, a sensitivity analysis was carried out. Each weight was modified by up to $\pm 0.1$, with a 0.05 step. In addition, the case where only one parameter was used is also considered, yielding 23 weight combinations as shown in Table 3.





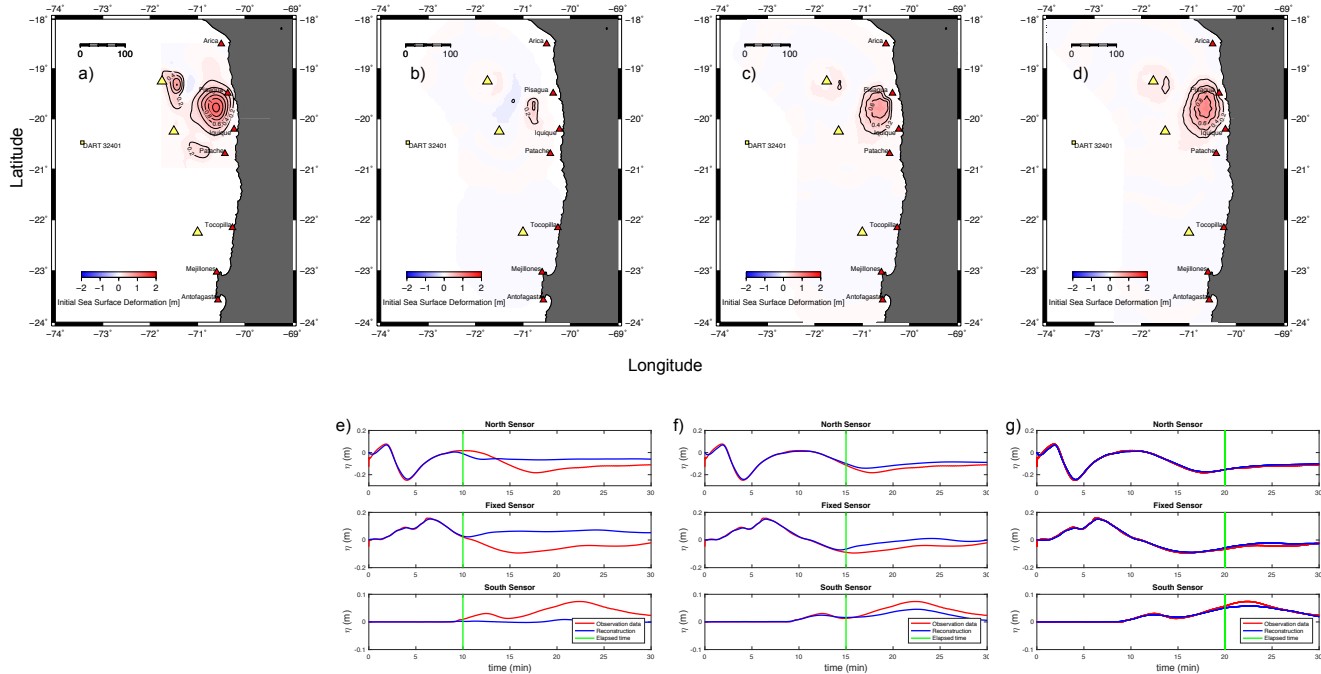

**Figure 7.** Sea surfaces obtained from different observation times for the 2014 Pisagua tsunami. (a) Faut model proposed by Hayes et al. (2014) used as target solution. (b) – (d) Distribution of the initial sea-surface displacement estimated by tsunami waveform inversion considering 10, 15 and 20 min of observation time respectively. (e) – (f) Comparison of observed (red lines) and calculated (blue lines) waveforms in the three sensors. Vertical green lines in (e), (f) and (g) indicate the data observation time.

As a first step, it is analyzed whether modifying the weights induces a change in the selection of the sensor array. In Fig 8a-d, sample spatial maps of the global error $EG$ are shown for some of the weight combinations (baseline combination, and combinations 10, 12 and 20, see Table 3 for details). It can be seen that although there is a variation of the value of the error, and also some variation of the spatial distribution of the magnitudes, the overall structure is consistent. The notable exception

is combination 20, which only considers $\alpha = 1.0$, thus considering only the error in arrival times. Fig. 8e shows the value of the global error $EG$ for each array configuration, as a function of the weight combination. For all combinations, the minimum error is obtained for the array 62, which is the one initially determined. This could be due to several factors. For instance, that the sensitivity range used did not suffice to alter the result. However, even when only one parameter is considered, the solution remains unaltered. Consequently, it could be possible that the solution is controlled by only one parameter. For instance, the

error in amplitude $\Delta H$ yields the minimum errors in several combinations (see reddish colors for combination 21 in Fig. 8e).





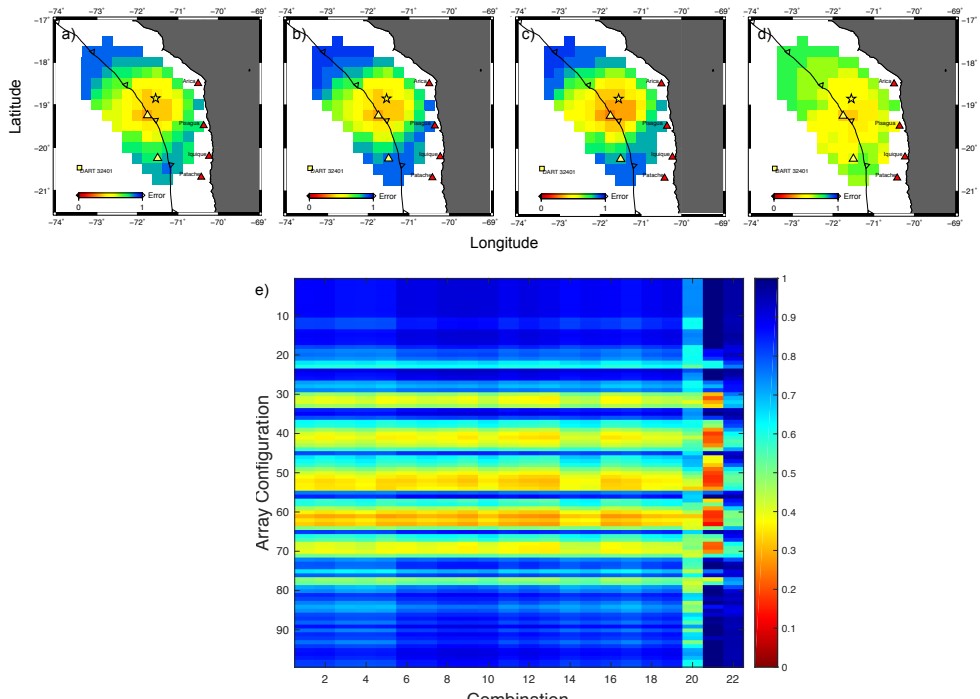

**Figure 8.** (a-d) Sample spatial maps of the global error, $EG$, as a function of the combination of weights. From left to right, combinations 1, 10, 12 and 20, respectively. (e) Matrix of the error $EG$ for each sensor array as a function of the combination of weights. Array configurations are numbered in accordance to the grid location, from west to east, and then from north to south.

**Table 2.** Summary of tsunami forecasting accuracy considering different data observation time for 2014 Pisagua event.

| | Observation time | | |
|---|---|---|---|
| Forecasting Point | 10 [min] | 15 [min] | 20 [min] |
| Arica | 0.66 | 1.00 | 1.00 |
| Piscagua | 0.83 | 0.90 | 0.70 |
| Iquique | 0.94 | 0.76 | 0.57 |
| Patache | 1.00 | 0.96 | 0.94 |
| Tocopilla | 0.83 | 0.77 | 0.79 |
| Mejillones | 0.59 | 0.55 | 0.54 |
| Antofagasta | 0.57 | 0.57 | 0.57 |





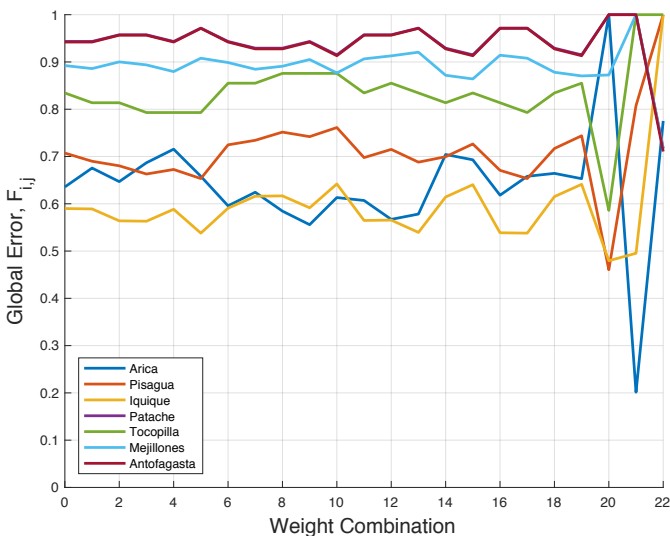

**Figure 9.** Forecasting accuracy by means of Equation 9 considering different weight combinations showed in the table 3 for the tsunami of Pisagua 2014.

However, there are some instances where it is the arrival time the parameter yielding the minimum error (see combination 20). Therefore, it is concluded that the use of a cost function combining both parameters maximizes the ability to capture well the overall structure of the tsunami signal. Moreover, the choice of weights proposed appears to be a good compromise among them.

Regarding the effect of the weighting on the accuracy assessment, it was also tested how much do the accuracy estimates for the Pisagua tsunami change with the different combinations of weights. The results are shown in Fig. 9, where it can be seen that in general, most forecast points have a relatively stable assessment, typically varying by less than 10% in the error estimate. However, the situation changes significantly if only one parameter is used. For instance, if only time is considered (combination 20), most forecast points worsen their assessment, as in Arica and Iquique. On the other hand, Patache sees a large drop in the error associated with the arrival time. The situation reverses when only skill is considered (combination 22), when Arica and Iquique improve the estimation but Patache worsens significantly. Therefore, it is found that the use of multiple parameters is essential in producing a more complete estimation of the error for determining the optimal configuration

The above results suggest the methodology is capable to deliver a working tsunami observation system comprised by just three sensors. Although the general rule that the spacing between sensor should be in the range ∼ 200-400 km (Bernard et al., 2001) is somewhat preserved, the spacing between sensors differs, between 110 and 225 km, approximately. This yields a





less dense network as the one proposed by either Schindelé et al. (2008) and Omira et al. (2009). Moreover, the methodology allows for the optimal placement of the sensors on this range of distances. On the other hand, the results show that, similarly to what is assumed by Gusman et al. (2014), the optimal placement corresponds to the area where the maximum displacement occurs. Considering the inherent uncertainty in predicting the slip distribution of the next earthquake, stochastic methods could

be incorporated to this methodology to further refine the results. However, with the simulations performed here, it is possible to see that a small number of sensors can provide a baseline solution.

However, at leas in a local context where arrival times are very short, it is also important to consider the actual benefit of incorporating such an evaluation in the context of early warning. The present results reaffirm that short observation times

yield less accurate results. The balance between a quick assessment and accuracy must be considered, and when the inherent uncertainties in inverting both earthquake and tsunami data are considered (Cienfuegos et al., 2018), it seems reasonable to propose that this information should be used for hazard assessment a later times, to refine quick hazard assessments estimated by other means, and not necessarily as the method for first evaluation..

## 6   Conclusions

This study presented a methodology, based on numerical simulations of near field tsunami forecasting, to determine an optimal array configuration of offshore tsunami sensors, with the goal of objectively determine the placement of the minimum set of sensors capable of providing an accurate estimation of the initial sea surface. To provide an objective basis for the comparison, three parameters are considered to assess the accuracy, thus expanding on previous methodologies that rely solely on arrival time. The joint use of the three estimators, arrival time $\Delta T$, tsunami amplitude $\Delta H$, and model skill $S_k$ was robust when

compared against a single parameter.

The methodology was tested in Northern Chile. Results showed that a configuration comprising a minimum of three sensors is capable of providing accurate estimations of the tsunami arrival time and peak amplitudes for the first wave. In this way, three sensors suffice to cover a 600 km stretch of coast when earthquakes in the $M_w$ 8.0 range are considered.

Results show that there is a strong dependency between the location of the sensors and error estimators; arrival times are accurately predicted with sensors located opposite to the coastal point of interest. In addition, better results for tsunami ampli-

**Table 3.** Parameter space of weighting functions used.

| Wei. | Combination | | | | | | | | | | | | | | | | | | | | | | |
|---|---|---|---|---|---|---|---|---|---|---|---|---|---|---|---|---|---|---|---|---|---|---|---|
| | 0 | 1 | 2 | 3 | 4 | 5 | 6 | 7 | 8 | 9 | 10 | 11 | 12 | 13 | 14 | 15 | 16 | 17 | 18 | 19 | 20 | 21 | 22 |
| $\alpha$ | 0.4 | 0.45 | 0.45 | 0.5 | 0.5 | 0.5 | 0.35 | 0.35 | 0.3 | 0.3 | 0.3 | 0.4 | 0.35 | 0.4 | 0.45 | 0.4 | 0.45 | 0.5 | 0.4 | 0.35 | 1.0 | 0 | 0 |
| $\beta$ | 0.4 | 0.35 | 0.4 | 0.35 | 0.3 | 0.4 | 0.45 | 0.4 | 0.45 | 0.5 | 0.4 | 0.45 | 0.5 | 0.5 | 0.3 | 0.3 | 0.45 | 0.4 | 0.35 | 0.35 | 0 | 1.0 | 0 |
| $\gamma$ | 0.2 | 0.2 | 0.15 | 0.15 | 0.2 | 0.1 | 0.2 | 0.25 | 0.25 | 0.2 | 0.3 | 0.15 | 0.15 | 0.1 | 0.25 | 0.3 | 0.1 | 0.1 | 0.25 | 0.3 | 0 | 0 | 1.0 |



tudes and skill are obtained when sensors are located inside the uplift or in front of it.

As a result, the methodology proposed shows promising potential to be used as an operational tool in defining the location of possible tsunameters, especially for sparse configurations in countries where the financial cost of implementing and maintaining

5 dense networks could be an insurmountable hurdle.

*Competing interests.*  The authors declare no competing interests/

*Acknowledgements.*  This work was developed within the "Research Project on ENhancement of Technology To Develop Tsunami-Resilient Community", sponsored by Science and Technology Research Partnership for Sustainable Development, SATREPS Program by Japan Science and Technology (JST), the Japan International Cooperation Agency (JICA). PAC would also like to thank CONICYT through its grants

10 FONDEF D11I1119 and IT15I10001, FONDAP 1511017 (CIGIDEN) and PIA-Basal FB0821 (CCTVal). Joaquin Meza has been supported by CONICYT through the MSc. Scholarship CONICYT-PCHA/Magister Nacional/2015–22150620 and by Dirección General de Investigación, Innovación y Postgrado, UTFSM.



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
