# Peer review of "A Methodology For Optimal Designing Of Monitoring Sensor Networks For Tsunami Inversion"

_Natural Hazards and Earth System Sciences, 2018_

## Referee Comment (RC1) · Anonymous Referee #1 · 19 Nov 2018

-A description of the inversion algorithm used to compute the initial ocean surface deformation from sensor-obtained time series should be provided. A great deal of the error and uncertainties the authors are trying to quantify have as much to do with limitations in the inversion scheme as with location of the sensors. Some of the observed errors may have little to do with sensor location, but a lot with other inversion parameters.

-Since the proposed sensor location is in the seismic rupture area (pre-seismic sensor depth will be different from post-seismic resulting in a BPR time series drift) an explanation of how the inversion process handles the drift in the sensor signal due to co-seismic displacement of the sea-floor should be provided.

-The authors should explain whether sensor signal noise (seismic (Raleigh waves),
tidal cycle, atmospheric pressure variations, electronics noise,...) has been included in the modeled sensor time series and justify why it was included or left out, and what the effects of inclusion or exclusion may be on inversion results.

-While the combination of three metrics into a single Accuracy Parameter is convenient for evaluation, it does not always result in the most accurate evaluation. For instance two identical time series with only a small phase shift originating from a small discrepancy in arrival time will result in very large error estimate in the "skill" (waveform) metric (as defined in Eq'n (5)), although both waveforms are basically identical, such wave forms should only show error in arrival time since H and 'skill' accuracy are 100%, but according to the metrics used, it will also sure large errors in the "skill". The authors should provide some comments on that or at least acknowledge the limitations of the metric used.

-The authors should explain the rationale for capping the error in all three metrics at 100%.

-A brief description of the JAGURS tsunami modeling code( mentioning physical model, discretization approach, validation tests) should be included in addition to the reference, for the benefit of the unfamiliar reader.

-pp8, l25: The authors state " 30 arc sec GEBCO global bathymetry is used since that resolution is sufficient for subfault size larger than 40x40 km", but in pp5, l10 they state the size of their finite faults is 700x700 arc sec ($\sim$20x20km), this seems to imply they will need higher resolution than 30 arc sec for the bathymetry. Perhaps, in their statement they mean "smaller than 40x40km", not larger?

pp5, l12-15: The authors state that due to their current locations outside the study area, DARTs will not be used as sensors. An explanation of why DARTs cannot be relocated if it is determined they would be more useful at different locations, would be appreciated.

-pp13,ll9: One of the conclusions states that the highest accuracy results are achieved when, at least, one sensor is located along the main energy beam. This is an obvious a priori conclusion, however it also points at the possibility that the identified locations may not be optimal for events with slip distributions disimilar to the test cases, particularly larger events with longer rupture lengths that may exhibit an energy beam not necessarily focused on the sensor. They authors should comment on that.

-The manuscript is for the most part well-written, however, some stylistic improvements and typos should still be reviewed and corrected. For instance: "...ar considered...", "...against an historical...", "...Hence, it highly desireable,...", "...range os sources...", "capable of constrain the source...", "it benefits for the fixed...", '...at leas in..."

---

## Referee Comment (RC2) · J. Behrens (Referee) · 20 Nov 2018

The article describes the development of a strategy to determine in a rational and reproducible way optimal positions for deep ocean tide gauges, such that tsunami early warning operations can be supported. The approach assumes a small number of deployable sensors and proposes a method that relies on an aggregate error measure, combined from different relevant quantities (arrival times, maximum wave height, and wave shape), to assess the forecast error. The result of this process is a list of optimal sensor locations.

While the report is generally well written and very relevant, and while the method proposed has some innovative potential, the manuscript still needs improvements before it

can be published as a full paper. I have compiled a number of remarks of more general character as well as specialized comments to specific parts of the text below.

Unfortunately, for practical applications the data quality of deep ocean wave gauges based on ocean bottom pressure sensors (OBPS) located close to the rupture area is almost useless. The large amplitude seismic waves (in particular Rayleigh waves) cannot be separated from the sea surface elevation signal close to the source. So, a clear wave form cannot be extracted from these sensors (the ones shown in figure 7 are highly unrealistic for sources so close to the wave gauge location). This caveat of the proposed method is not discussed nor considered. This principle drawback was the major reason, why the buoys were abandoned from the German-Indonesian Tsunami Early Warning System (besides considerations regarding maintenance).

**General Remarks**

1. While the data quality obtainable by coastal gauges is discussed (and discarded) the quality of deep ocean sensors (in particular OBPS) is highly dependent on the proximity to the source. If short lead times are an issue then these sensors in practice do not provide accurate information, unless appropriately filtered (which is impossible for many such situations). However, in the methodology chapter, it is assumed that the free surface information can be retrieved with sufficient accuracy, independent of the sensor location. This is an assumption which cannot be fulfiled in practice. Please discuss this issue and consider it in the optimization approach.

2. Using a Green's function approach can be justified only, if the underlying process is (almost) linear. While far-field tsunami propagation can be approximated very well by a linear approximation, I doubt that is also possible in the near field, where wave interferences may lead to highly non-linear effects.

3. I do not understand what you mean with the listening area and placing sensors

within that same. Maybe a sketch would help?

4. I also do not clearly understand what metric you use (you call it error and it is depicted in figures 2, and 3) to assess the forecast quality. As far as I understand, you compute the (virtual) reality by using a forward simulation from the given source by COMCOT. Then you use the Green's function approach (computed by JAGURS) to invert from the COMCOT reality to an inverted source, if I am not mistaken. But then, what do you use to compare? Do you use that inverted source run yet another forward simulation and compare? You should probably describe this approach more clearly.

5. Since only two major scenarios are used for the sensor positioning and the optimal sensor locations are then close to the two epicenters, it should be evaluated how the proposed optimal positions depend on the choice of scenarios.

**Specific Comments**

1. P1, L14: ar → are

2. P2, L25: I think it would be fair to include the following reference (sorry for the self promotion, but it is relevant here): J. BEHRENS, A. ANDROSOV, A. Y. BABEYKO, S. HARIG, F. KLASCHKA, L. MENTRUP (2010): A new multi-sensor approach to simulation assisted tsunami early warning, Nat. Hazards Earth Syst. Sci., 10:1085-1100, DOI:10.5194/nhess-10-1085-2010.

3. P2, L25: Hence it highly → Hence it is highly

4. P3, L7: In the case of Japan, they have... - consider to change the sentence for a better style

5. P5, L7: Greeen's → Green's

6. P5, L29: $T_1$ is indeed used in a TWS, i.e. InaTEWS (see Rakowksy et al. (2013): Operational tsunami modelling with TsunAWI – recent developments and applications, Nat. Hazards Earth Syst. Sci., 13:1629-1642, DOI:10.5194/nhess-13-1629-2013.)

7. P6, Eq. (6)-(8): Why do you take the minimum with 1 here, this is no normalization! Dividing through the maximum would normalize, maybe... But this is just a "limiting". Additionally, it does not preserve the specific behavior of the corresponding functions ($H$, $T_m$, $S_k$). You argue that a limit of 1 indicates 100% relative error, which does make sense. But you should clarify this: If the values are normalized then they cannot exceed 1.

8. P7, Eq. (9): It is important to stress that the sum of weights needs to be 1 again in order to preserve the normalization! You did this intuitively, but should point this out.

9. P7, Eq. (10): I am not sure if you want to sum up all values. This will give you a non-normalized norm. Additionally, by summing the values up, this corresponds to a 1-norm. Why didn't you take a 2-norm (squareroot of summed squared values)? Why didn't you devide by the number of forecast points $N$ to get a normalized value again?

10. P7, L21: Why do you use a different notation to Eq. 10 here? ($F_{ij}(\cdots)$ vs. $F(\cdots)_{ij}$?)

11. P10, L15: assumemption → assumption

12. P10, L23ff.: That the use of different tsunami models for producing Green's functions and for propagating to forecast points can be considered as replication of natural variability (aka uncertainty) appears quite ad hoc. You should probably justify this by looking at some quantitative measures.

13. P18, Table 2: Piscagua → Pisagua

14. P19, Figure 9: the line for Patache is not visible

15. P20, L8: at leas → at least

---

## Author Comment (AC1) · 26 Jan 2019

A description of the inversion algorithm used to compute the initial ocean surface deformation from sensor-obtained time series should be provided. A great deal of the error and uncertainties the authors are trying to quantify have as much to do with limitations in the inversion scheme as with location of the sensors. Some of the observed errors may have little to do with sensor location, but a lot with other inversion parameters.

1. Since the proposed sensor location is in the seismic rupture area (pre-seismic sensor depth will be different from post-seismic resulting in a BPR time series drift) an explanation of how the inversion process handles the drift in the sensor signal due to co-seismic displacement of the sea-floor should be provided.

[Figure]

R:> It is true that inversion algorithms may be subject to inaccuracies arising from a wide range of origins and limitations. However, in the context of the present work, it is assumed that these inaccuracies will weigh equally for all tested inversions since the process is the same and, consequently, the difference in result can be attributed to sensor placement. In that sense, it is an underlying hypothesis that the methodology as proposed can be used with other inversion schemes as well.

In particular, the algorithm used in this article was proposed by Tsushima et al (2012) (op.cit. in the manuscript). They proposed a method of tsunami waveform inversion to accurately estimate a tsunami source by incorporating the effect of permanent seafloor deformation recorded by ocean-bottom pressure gauges (OBPGs) within the source region. In order to take the drift into consideration, it is assumed that coseismic rupture propagates with infinite velocity. The general expression is reduced to an equation relating observed OBPG waveforms to initial sea-surface displacement at the source by using a Green's function consisting of two terms: The Green's function used in regular tsunami inversion and a correction term to account for water-depth change in response to permanent seafloor deformation.

We have modified the text to highlight the hypothesis mentioned above and that the use of this particular inversion scheme is aimed to be a proof-of-concept rather than relying specifically on this method. (See the expanded Page 4, line 27-33).

Page 4, line 27-33: "The method builds on the premise that, in order to determine the tsunami source, an inversion procedure must be implemented. While an inversion algorithm can be subject to a wide range of errors and uncertainties on its own, for the purpose of the present work it is hypothesized that using a single inversion procedure will weigh equally those errors allowing comparison 30 among different sensor configurations. Consequently, it is the procedure presented herein can be applied independently of the inversion method, and the nature of the sensors. Similarly, other sources of signal noise such as atmospheric variations, tides, etc."

2. The authors should explain whether sensor signal noise (seismic (Raleigh waves), tidal cycle, atmospheric pressure variations, electronics noise,...) has been included in the modeled sensor time series and justify why it was included or left out, and what the effects of inclusion or exclusion may be on inversion results.

R:> It is true that in the original version of the manuscript, these effects were not mentioned. We follow previous research in assuming that pressure variations due to non tsunami phenomena (ocean tides, atmospheric and oceanographic disturbances, vertical deformation of the seafloor, and seismic waves) have very different time scales from those of tsunamis and can therefore be easily separated, e.g. Tsushima et al (2009). For operational settings, Tsushima et al (2009, 2012) applied a 60-s moving average and a low-pass digital filter with a cutoff period of 60 s (Saito, 1978) to remove seismic waves (acoustic waves which are converted from body waves at the seafloor). The ocean tide components were removed by subtracting the sea level variation computed from a theoretical tide model developed by Matsumoto et al. (2000). Those subtraction was implemented during realtime waveform data processing. For our purposes, however, we did not consider these effects and even in the case of the Illapel earthquake and tsunami, the tides were removed during preprocessing.

We have mentioned our scope now in the text, Page 4, L31-33.

Page 4, L31-33: "Similarly, other sources of signal noise such as atmospheric variations, tides, etc. It is noted also that in actual operational conditions, incoming data from sensors could need to be pre-processed as to improve the signal to noise ratio. For instance, Tsushima et al. (2012) use a 60 s moving average to process data from ocean bottom pressure sensors."

3. While the combination of three metrics into a single Accuracy Parameter is convenient for evaluation, it does not always result in the most accurate evaluation. For instance two identical time series with only a small phase shift originating from a small discrepancy in arrival time will result in very large error estimate in the "skill" (waveform) metric (as defined in Eq'n (5)), although both waveforms are basically identical, such wave forms should only show error in arrival time since H and 'skill' accuracy are 100%, but according to the metrics used, it will also sure large errors in the "skill". The authors should provide some comments on that or at least acknowledge the limitations of the metric used.

R:> We think we understand the source of the reviewers' concerns, but we think the example highlights the benefits of combining multiple metrics. Of the two time series alluded (note we understand these are two time series resulting from the inversion, not two time series that need to be compared one against the other directly as would be the case if one of them is the target and the other the observed time series), it is expected that only one of them would be closer to the true series. Hence, one of them will show better values in arrival time and skill, while sharing the same value for height. Hence, one of them will be evaluated as better among the two but probably would compare less favorably against another having better skill.

We have added a short introduction explaining the idea behind the combined metric, in page 6, L24 and page 7, L1-14.

4. The authors should explain the rationale for capping the error in all three metrics at 100%.

R:> The original intention was to provide a common base of comparison, and to minimize the possibility of one of the metric carrying too much weight in the evaluation. By saturating at 100%, all cases reaching this value are weighted the same thereby allowing discrimination by differences in the other metrics.

We had briefly mentioned this in the original manuscript, but we have now expanded the description in page 7, line 4-19

Page 7, line 4-10: "In assessing the accuracy, the error between observed and forecasted quantities is estimated for arrival times and maximum 5 tsunami amplitudes.

[Figure]

However, one possible difficulty in establishing a standard metric is that each of these parameters has its own scale with significantly different ranges of values. For example, while a error in arrival time of a a few minutes can be considered reasonable (for instance, less than five), a variation in height of more than one meter can signify a large error. To provide a common comparison basis for all possible sensor configurations, each parameter error is adimensionalized by dividing by the reference provided by the observed dataset. In addition, it is possible that one parameter having a large error 10 could bias the combined assessment to be implemented. Therefore, the error estimated is capped under the consideration that errors larger the observed value will be treated as equally significant.

5. A brief description of the JAGURS tsunami modeling code (mentioning physical model, discretization approach, validation tests) should be included in addition to the reference, for the benefit of the unfamiliar reader.

R:> The suggestion was considered, and is included in Page 9, Line 14-17.

Page 9, Line 14-17: "JAGURS is a numerical model for dispersive tsunami wave modeling. This is a parallel software which solves the nonlinear Boussinesq dispersive equations in spherical coordinates. To solve these equations, a leapfrog staggered-grid, finite-difference calculation scheme is used."

6. pp8, l25: The authors state " 30 arc sec GEBCO global bathymetry is used since that resolution is sufficient for subfault size larger than 40x40 km", but in pp5, l10 they state the size of their finite faults is 700x700 arc sec (_20x20km), this seems to imply they will need higher resolution than 30 arc sec for the bathymetry. Perhaps, in their statement they mean "smaller than 40x40km", not larger?

R:> We have decided to remove the sentence, to avoid confusion. Our minimum source size would be constrained by a single source and the grid used is 23 times smaller than the unit source size along one dimension.

7. pp5, l12-15: The authors state that due to their current locations outside the study area, DARTs will not be used as sensors. An explanation of why DARTs cannot be relocated if it is determined they would be more useful at different locations, would be appreciated.

R:> As stated in previous responses, the goal of the article is to provide a methodology to compare and evaluate among different sensor configurations. Whether these configurations include or not DART sensors are not intrinsically relevant, although they could easily be included if needed. Moreover, the basis is to evaluate how to design a new sensor array, for instance, in places where a system was to be built from scratch. A byproduct of this work could be the relocation of a DART buoy, but it is not the intended goal to be site-specific.

We acknowledge that the original wording in the article may induce confusion from this intended meaning, therefore we have modified the text to clarify this. See page 5, Line 20-25.

P page 5, Line 20-25 :"They cover an area spanning about 7âŮę ×3âŮę (latitude and longitude), consistent with the so called Northern Chile Gap (Comte and Pardo, 1991; Metois et al., 2013). In what follows, although an inversion process can benefit by other data sources such as Deep ocean Assessment and Reporting of Tsunamis (DART) buoys, these are not considered in the analysis under the premise that the area of interest is developing a completely new system. On the other hand, for the particular case of Chile, the location of the existing DART buoys 25 in the area of interest is such that requires longer observation times than the ones studied here (Williamson and Newman, 2018)."

8. pp13,ll9: One of the conclusions states that the highest accuracy results are achieved when, at least, one sensor is located along the main energy beam. This is an obvious a priori conclusion, however it also points at the possibility that the identified locations may not be optimal for events with slip distributions disimilar to the test

cases, particularly larger events with longer rupture lengths that may exhibit an energy beam not necessarily focused on the sensor. They authors should comment on that.

R:> It is true that the performance would degrade for actual cases where the slip distribution would be such that no sensors are located near the main energy beam. Acknowledging this, we evaluated scenarios where the target earthquake was located in between the original target earthquakes, and also tested the performance for an existing earthquake with its slip distribution. While some degradation of the results is present (see Figures 5-7), the overall performance is considered acceptable.

We have now updated the discussion on this regard in the updated manuscript. See P18, 2-19.

9. The manuscript is for the most part well-written, however, some stylistic improvements and typos should still be reviewed and corrected. For instance:

"...ar considered..."

"...against an historical..."

"...Hence, it highly desireable,..."

"...range os sources..."

"capable of constrain the source..."

"it benefits for the fixed..."

"...at leas in..."

R:> Thank you for catching these typos. The suggestions were considered. However, we noticed that these were not all instances. Therefore, these and some other modifications in the text aimed at improving style and grammar were introduced. The reviewer is referred to the version with tracked changes for details.

Please also note the supplement to this comment:
https://www.nat-hazards-earth-syst-sci-discuss.net/nhess-2018-269/nhess-2018-269-AC1-supplement.pdf
* * *

---

## Author Comment (AC2) · 26 Jan 2019

The article describes the development of a strategy to determine in a rational and reproducible way optimal positions for deep ocean tide gauges, such that tsunami early warning operations can be supported. The approach assumes a small number of deployable sensors and proposes a method that relies on an aggregate error measure, combined from different relevant quantities (arrival times, maximum wave height, and wave shape), to assess the forecast error. The result of this process is a list of optimal sensor locations.

While the report is generally well written and very relevant, and while the method proposed has some innovative potential, the manuscript still needs improvements before it

can be published as a full paper. I have compiled a number of remarks of more general character as well as specialized comments to specific parts of the text below. Unfortunately, for practical applications the data quality of deep ocean wave gauges based on ocean bottom pressure sensors (OBPS) located close to the rupture area is almost useless. The large amplitude seismic waves (in particular Rayleigh waves) cannot be separated from the sea surface elevation signal close to the source. So, a clear wave form cannot be extracted from these sensors (the ones shown in figure 7 are highly unrealistic for sources so close to the wave gauge location). This caveat of the proposed method is not discussed nor considered. This principle drawback was the major reason, why the buoys were abandoned from the German-Indonesian Tsunami Early Warning System (besides considerations regarding maintenance).

R:> Thank you for the comments. Indeed, there are some operational aspects that further constrain locations. We did not intend to address these in the present manuscript, as they can be sensor-specific. Instead, we note that if such constraints are well understood, the possible location of the sensors can be restricted, but the methodology and base of comparison can be used as presented. We have now explicitly mentioned this. See Page 4, L27-33, P9; L21-24.

Page 4, Line 27-33: "The method builds on the premise that, in order to determine the tsunami source, an inversion procedure must be implemented. While an inversion algorithm can be subject to a wide range of errors and uncertainties on its own, for the purpose of the present work it is hypothesized that using a single inversion procedure will weigh equally those errors allowing comparison 30 among different sensor configurations. Consequently, it is the procedure presented herein can be applied independently of the inversion method, and the nature of the sensors. Similarly, other sources of signal noise such as atmospheric variations, tides, etc. It is noted also that in actual operational conditions, incoming data from sensors could need to be pre-processed as to improve the signal to noise ratio.

Page 9, Line 21-24: Although in operational conditions, the time series recorded at an

actual sensor will also include other signals that could be considered as tsunami noise, these are not considered here under the premise that the approach used herein is to discriminate among sensor configurations, assuming the treatment of tsunami noise is equivalent across sensors. Similarly, it is noted that possible operational restrictions regarding sensor placement are obviated although they could be implemented straightforwardly."

General Remarks

1. While the data quality obtainable by coastal gauges is discussed (and discarded) the quality of deep ocean sensors (in particular OBPS) is highly dependent on the proximity to the source. If short lead times are an issue then these sensors in practice do not provide accurate information, unless appropriately filtered (which is impossible for many such situations). However, in the methodology chapter, it is assumed that the free surface information can be retrieved with sufficient accuracy, independent of the sensor location. This is an assumption which cannot be fulfilled in practice. Please discuss this issue and consider it in the optimization approach.

R:> Effectively, sensor placement can be subject to constraints which have not been considered here, such as the ones mentioned. However, imposing such constraints should be straightforward and do not affect the overall methodology as presented. On the other hand, spatial dependencies are weighed equally because, at this stage, it is considered that the network would work at a wide range of possible sources. Inclusion of these effects could be subject of subsequent work.

For completeness, we have included a comment on this regard in the discussion. See for instance P4-30- P5L4, and in the Conclusions.

Page 4, Line 30 to page 5, line 4: "For instance, Tsushima et al. (2012) use a 60 s moving average to process data from ocean bottom pressure sensors. Moreover, in some cases, the quality of the data could be dependent on distance to the source. For the present case, these effects are not considered and it is assumed the inversion

process is free of its influence. In addition, it is expected that the resulting network could be used for a range of source locations, hence spatial dependencies are not considered and weighed equally."

2. Using a Green's function approach can be justified only, if the underlying process is (almost) linear. While far-field tsunami propagation can be approximated very well by a linear approximation, I doubt that is also possible in the near field, where wave interferences may lead to highly non-linear effects.

R:> It is true that nonlinear processes, complex interaction with bathymetry and coastal morphology can lead to nonlinear response that could prevent proper characterization of the source. Moreover, the use of coastal gages in inversion can be subject to inverting for coastal effects and trapping. However, it is common practice to use only the first few minutes of the incoming tsunami to minimize these effects. In the present case, we have considered this by placing sensors in deeper water (typically deeper than 50 m) and by considering only up to 30 min of elapsed time to be used in the inversion. Therefore, regarding the inversion, nonlinear effects might not be an issue. See comment on L29-31, pp 5.

On the other hand, we used the coastal points to assess the accuracy of the forecast once a source was found by the inversion. In this case, the role of nonlinearity and other effects could be more relevant, and it could depend on the characteristics of each inverted source. One way to address this would have needed to run the complete, forward simulation including nonlinearity, but it would have been too costly since a solution would have been needed for each sensor array. However, it is expected that omitting these processes will affect the accuracy but not does limit discriminating among sensor configurations. Moreover, coastal points are located at 200 m. depths to reduce the effect of nonlinearity.

We have now acknowledged this aspect and added it to the manuscript. See L30 and following, pp11.

Page 5, Line 29-31: "Typically, these are located at absolute depths such than nonlinear 30 effects can be neglected thereby linear superposition of tsunami time series can be performed with minimal errors. The coastal observation points correspond to tide gages, which are used to evaluate the predictive performance at the coast"

Page 11, line 31-35: "It is of note that the use of linear superposition at shallow coastal points might be subject to inaccuracies arising from neglecting nonlinear interactions, and other process such as bathymetry–induced effects such as resonance. However, it is expected that omitting these process will affect the accuracy but not does limit discriminating among sensor configurations. Moreover, coastal points are located at 200 m. depths to reduce the effect of nonlinearity."

3. I do not understand what you mean with the listening area and placing sensors within that same. Maybe a sketch would help?

R:> The listening area is meant to be the space enclosed by the tsunami wavefront at any given time. (Tsushima et al., 2009, op.cit. denotes it "influence area") It allows determining which sensors would be able to record some of the tsunami signal for a given elapsed time since triggering. It is determined by computing the forward-propagated wavefronts from each tsunami event using ray tracing.

We have attempted to clarify the wording in the update version of the manuscript. See paragraph L18-23, pp8.

Page 8, Line 18-23: "On the other hand, the relative distance of the sensors will depend on the time allowed to record the tsunami (e.g. Bernard et al., 2001). Hence, given an earthquake, it is possible to define the area (henceforth termed listening area Al) over which the tsunami waves have already propagated away from the source(e.g. Williamson and Newman, 2018). A minimum of two sensors must be considered inside this area. This equivalent to defining a data observation time, if the tsunami propagation speed is known. This time, termed T0, represents the time the TWS allows for recording of the tsunami before performing an inversion."

4. I also do not clearly understand what metric you use (you call it error and it is depicted in figures 2, and 3) to assess the forecast quality. As far as I understand, you compute the (virtual) reality by using a forward simulation from the given source by COMCOT. Then you use the Green's function approach (computed by JAGURS) to invert from the COMCOT reality to an inverted source, if I am not mistaken. But then, what do you use to compare? Do you use that inverted source run yet another forward simulation and compare? You should probably describe this approach more clearly.

R:> The procedure is as described by the reviewer, with the exception that a linear combination of the unit sources (computed by JAGURS) are used to estimate the "forecast solution" at each tidegage, where they are compared with the COMCOT-modeled full forward model.

We have now updated the description on this regard in the updated manuscript to clarify it further. See L21-35, pp11. We have also updated the caption of Figures 2, 3

Page 11, Line 11-35: To carry out the analysis, each scenario is propagated forward to all forecast points (FPs), using the Nonlinear Shallow Water Equations as implemented in the model COMCOT. The time series at nodes in deep water are considered as target (observed) time series to be used in the inversion process, whereas the use of coastal forecast points is aimed at establishing the quality of the assessment (see below). The use of a different tsunami model in propagating the signal and in preparing the database of Green's functions allows for differences in the target tsunami time series to those of the database, to reduce possible overfitting in the inversion. No other source of variability, such as noise, is considered. It is also assumed that tsunami-tide interactions do not play a significant role in this area.

For each of the 212 sensor arrays (99 and 113, for each scenario), these observed tsunami time series are used to invert the tsunami source. Once a source is determined, the linear combinations of the Green's functions consistent with the source weights are used to estimate the forecast time series in the coastal points. It is of

note that the use of linear superposition at shallow coastal points might be subject to inaccuracies arising from neglecting nonlinear interactions, and other process such as bathymetry–induced effects such as resonance. However, it is expected that omitting these process will affect the accuracy but not does limit discriminating among sensor configurations. Moreover, coastal points are located at 200 m. depths to reduce the effect of nonlinearity"

5. Since only two major scenarios are used for the sensor positioning and the optimal sensor locations are then close to the two epicenters, it should be evaluated how the proposed optimal positions depend on the choice of scenarios.

R:> It is true that the performance would degrade for actual cases where the slip distribution would be such that no sensors are located near the main energy beam or away from the original two epicenters. Acknowledging this, we evaluated scenarios where the target earthquake was located in between the original target earthquakes, and also tested the performance for an existing earthquake with non uniform slip distribution.

We note the sensitivity to the scenario choice has not been tested. For instance, what would have been the case if a Mw7.5 target scenario would have been used instead. However, the method would allow for a solution when other scenarios are used. One can speculate they will be different to the one obtained here. Determining which one is best, would be debatable. Here, we have attempted to find a configuration that works for a certain type of scenario, mostly controlled by the earthquake magnitude.

We have now acknowledged this in the discussion on this regard in the updated manuscript. See P18, L7-10

Page 18, Line 12-16, "This suggests that the proposed network is capable of identifying smaller events with non-uniform slip reasonably well, in reasonable time. It is of note also that the location of this scenario does not coincide with either of the scenarios used to design the network. However, two of the sensors are located close but not directly into the main energy beam. When considered in 15 unison, these two tests suggest

that the sensor network considered would be appropriate for tsunamis generated by earthquakes of magnitude similar and larger than that of the initial target scenarios"

Specific Comments

1. P1, L14: ar -> are

R:> The suggestion was considered.

Page 1, Line 14 - ar -> are

2. P2, L25: I think it would be fair to include the following reference (sorry for the self promotion, but it is relevant here): J. BEHRENS, A. ANDROSOV, A. Y. BABEYKO, S. HARIG, F. KLASCHKA, L. MENTRUP (2010): A new multi-sensor approach to simulation assisted tsunami early warning, Nat. Hazards Earth Syst. Sci.,10:1085-1100, DOI:10.5194/nhess-10-1085-2010.

R:> The suggestion was considered.

Page 2, Line 25 - estimating the tsunami hazard (e.g. Gusman and Tanioka, 2014; Gusman et al., 2014; Cienfuegos et al., 2018). -> estimating the tsunami hazard (e.g. Behrens et al., 2010; Gusman and Tanioka, 2014; Gusman et al., 2014; Cienfuegos et al., 2018).

3. P2, L25: Hence it highly -> Hence it is highly

R:> The suggestion was considered.

Page 2, Line 25 - Hence it highly -> Hence it is highly

4. P3, L7: In the case of Japan, they have... - consider to change the sentence for a better style

R:> The suggestion was considered.

Page 3, Line 7 - In the case of Japan, they have -> For instance, Japan has a few submarine

5. P5, L7: Greeen's -> Green's

R:> The suggestion was considered.

Page 5, Line 7 - Greeen's -> Green's

6. P5, L29: T1 is indeed used in a TWS, i.e. InaTEWS (see Rakowksy et al. (2013): Operational tsunami modelling with TsunAWI – recent developments and applications, Nat. Hazards Earth Syst. Sci., 13:1629-1642, DOI:10.5194/nhess-13-1629-2013.)

R:> The reference was considered.

Page 5, Line 29 - "Although this definition is not used in operational TWS, it is considered a relevant estimate for the time required to trigger 30 a warning status. However, considering that in some cases, the tsunami time series might not exceed the threshold" -> This arrival time definition is also used by the German-Indonesian Tsunami Early Warning System (GITEWS) (Rakowsky et al 2013)

7. P6, Eq. (6)-(8): Why do you take the minimum with 1 here, this is no normalization! Dividing through the maximum would normalize, maybe... But this is just a "limiting". Additionally, it does not preserve the specific behavior of the corresponding functions (H, Tm, Sk). You argue that a limit of 1 indicates 100% relative error, which does make sense. But you should clarify this: If the values are normalized then they cannot exceed 1.

R> Indeed, it is effectively capped at a certain value. We have changed the word "normalized" by "adimensionalized".

The original intention was to provide a common base of comparison, and to minimize the possibility of one of the metric carrying too much weight in the evaluation. By saturating at 100%, all cases reaching this value are weighted the same thereby allowing discrimination by differences in the other metrics.

We had briefly mentioned this in the original manuscript, but we have now expanded

the description in line 8-9, page 7.

Page 7, Line8-9: "To provide a common comparison basis for all possible sensor configurations, each parameter error is adimensionalized by dividing by the reference provided by the observed dataset."

8. P7, Eq. (9): It is important to stress that the sum of weights needs to be 1 again in order to preserve the normalization! You did this intuitively, but should point this out.

R:> Thank you for pointing out the omission. Indeed, we considered it implicitly but did not declare it. We have pointed this explicitly in P7, L26-27.

Page 7, Line 26-27: "The sum of the weights should add to one, as to preserve the comparison basis."

9. P7, Eq. (10): I am not sure if you want to sum up all values. This will give you a non-normalized norm. Additionally, by summing the values up, this corresponds to a 1-norm. Why didn't you take a 2-norm (squareroot of summed squared values)? Why didn't you devide by the number of forecast points N to get a normalized value again?

R:> It was a typo, as we had considered dividing by the number of forecast points.

10. P7, L21: Why do you use a different notation to Eq. 10 here? ($F_{ij}( . . .)$ vs. $F( . . .)_{ij}$?)

R:> The suggestion was considered.

Page 7, Line 21 - $F( . . .)_{ij}$ -> $F_{ij}( . . .)$

11. P10, L15: assumemption -> assumption

R:> The suggestion was considered.

Page 10, Line 15 - assumemption -> assumption

12. P10, L23ff.: That the use of different tsunami models for producing Green's functions and for propagating to forecast points can be considered as replication of natural

variability (aka uncertainty) appears quite ad hoc. You should probably justify this by looking at some quantitative measures.

R:> Rather than preserving natural variability, our intention was for minimize possible overfitting that could occur when the input and target signals are created with the same model. We have reworded the text to clarify this.

13. P18, Table 2: Piscagua -> Pisagua

R:> The suggestion was considered

Page 18 - Table 2 - Piscagua -> Pisagua

14. P19, Figure 9: the line for Patache is not visible

R:> Patache is not visible since Patache and Antofagasta are overlapped. They have a similar error, close to the unit. Therefore, if weights are changed, the behavior is similar for all evaluated combination.

Page 20, Line 10 - On the other hand, Patache and Antofagasta, which are overlapped since they have a similar, but not equal error, close to the saturation limit, sees a large drop in the error associated with the arrival time.

15. P20, L8: at leas -> at least

R:> Thank you for catching these typos. The suggestions were considered. However, we noticed that these were not all instances. Therefore, these and some other modifications in the text aimed at improving style and grammar were introduced. The reviewer is referred to the version with tracked changes for details.

Please also note the supplement to this comment:
https://www.nat-hazards-earth-syst-sci-discuss.net/nhess-2018-269/nhess-2018-269-AC2-supplement.pdf

**Supplement:**

[revised manuscript text omitted]
. While an inversion algorithm can be subject to a wide range of errors and uncertainties on its own, for the purpose of the present work it is hypothesized that using a single inversion procedure will weigh equally those errors allowing comparison among different sensor configurations. Consequently, it is the procedure presented herein can be applied independently of the inversion method, and the nature of the sensors. Similarly, other sources of signal noise such as atmospheric variations, tides, etc. It is noted also that in actual operational conditions, incoming data from sensors could need to be pre-processed as to improve the signal to noise ratio. For instance, Tsushima et al. (2012) use a 60 s moving average to process data from ocean

bottom pressure sensors. Moreover, in some cases, the quality of the data could be dependent on distance to the source. For the present case, these effects are not considered and it is assumed the inversion process is free of its influence. In addition, it is expected that the resulting network could be used for a range of source locations, hence spatial dependencies are not considered and weighed equally.

In this case, the "tsunami Forecasting based on Inversion for initial sea-Surface Height" (tFISH) algorithm is used (see Tsushima et al., 2009, for details). To carry out the inversion analysis, offshore tsunami waveform data are inverted (in real time in an operational setting) to estimate the initial distribution of the sea surface displacement. Coastal tsunami waveforms are then synthesized by linearly combining pre-computed tsunami Green's functions weighted by the resulting initial distri-

10  bution. The method allows for considering the coseismic displacement in the inversion procedure, thereby allowing to place sensors in the seismogenic zone. More details about tFISH can be seen in Tsushima et al. (2009, 2012).

The use of Green's functions is preferred over directly modeling individual events to target sensor locations (e.g. Schindelé et al., 2008) because it allows for testing and comparing a large number of sensor configurations at low computational cost,

15  provided a quantitative parameter or cost function is defined for the comparison. To compute the Green's Function for each subfault patch, a numerical scheme of the linear long-wave equations is used to propagate each tsunami elemental source to a set of predefined forecast points. The initial sea surface model for each Green's function is represented by a Gaussian function. Overlapping Green's functions are considered to express smooth variations of sea-surface displacement with a finite number of discrete elements (Aida, 1972). For the purposes of the present work, a set of nearly 1000 Green's functions is considered.

20  Each of them is of dimensions of 700 x 700 arc seconds, with its centers spaced 0.15 arc-deg. They cover an area spanning about $7° \times 3°$ (latitude and longitude), consistent with the so called Northern Chile Gap (Comte and Pardo, 1991; Metois et al., 2013). In what follows, although an inversion process can benefit by other data sources such as Deep ocean Assessment and Reporting of Tsunamis (DART) buoys, these are not considered in the analysis under the premise that the area of interest is developing a completely new system. On the other hand, for the particular case of Chile, the location of the existing DART buoys

25  in the area of interest is such that requires longer observation times than the ones studied here (Williamson and Newman, 2018).

Tsunami free surface time series, $\eta(t)$, are computed at a set of offshore and coastal observation points for a prescribed duration of the event, $T$. The offshore observation points are considered possible locations of a single observing station or sensor, and a subset of sensors is termed a sensor array. Typically, these are located at absolute depths such than nonlinear

30  effects can be neglected thereby linear superposition of tsunami time series can be performed with minimal errors. The coastal observation points correspond to tide gages, which are used to evaluate the predictive performance at the coast.

The overall procedure considers that given a source event, and for a given observation array, it is possible to estimate an inverted tsunami source solution. This solution is then propagated to the coastal points, where a set of relevant parameters are

35  assessed to evaluate the quality of the solution by comparing it against a target scenario calculated independently. To this end,

the error in three tsunami parameters is estimated. As others (e.g. Schindelé et al., 2008; Omira et al., 2009), tsunami arrival time is considered a key parameter, as it is essential in a TWS framework to provide a timely hazard assessment.

The definition of arrival time is relatively loose and could refer to different stages of the tsunami, such as the first exceedance of a threshold, the first local crest, first initial trough of N-waves, and others (e.g. Hayashi et al., 2011). Here, two different definitions are considered in order to make the analysis more robust. The first definition defines the time $T_1$ at which the tsunami exceeds a certain threshold

$$T_1(\eta) = \min(t \in (0,T) \mid \eta(t) > \epsilon). \tag{1}$$

This arrival time definition is also used by the German-Indonesian Tsunami Early Warning System (GITEWS) (Rakowsky et al., 2013). However, considering that in some cases, the tsunami time series might not exceed the threshold, a second arrival time is defined as the time when a proxy for the slope of the free surface exceeds a certain threshold

$$T_2(\eta) = \min \left( t \in (0,T) \mid \frac{\partial \eta}{\partial t} > \delta \right). \tag{2}$$

In this case, the idea is to establish a measure of the rate of change of the tsunami signal as an early proxy for the first local maximum. In doing this, it is assumed that the actual free surface slope is proportional to the rate of change as per the long wave approximation

$$\frac{\partial \eta}{\partial x} \propto \frac{\partial \eta}{\partial t}. \tag{3}$$

However, the accuracy in predicting the free surface time series is also relevant for a TWS. A key parameter is the maximum tsunami amplitude (here denoted as $H$), which is the parameter used to categorize the hazard in most existing TWSs. The maximum tsunami amplitude is estimated as

$$H = \max(\eta(t) - \eta(0) \mid t \in (0,T)). \tag{4}$$

It is of note that neither arrival time nor the maximum tsunami amplitude consider the accuracy in retrieving the shape of the waveform. Hence, the skill estimator is also introduced ($S_k$), as this index is commonly used to evaluate numerical performance of a model (e.g. Hampson et al., 2011)

$$S_k = \sqrt{\frac{\frac{1}{n} \sum_{i=1}^{n} \left( \eta_{\text{for}} - \eta_{obs} \right)^2}{\frac{1}{n} \sum_{i=1}^{n} \left( \eta_{obs} \right)^2}}, \tag{5}$$

where $\eta_{\text{for}}$ and $\eta_{obs}$ are the time series forecasted from the inverted source, and the target or observed source at a location of interest, respectively. $n$ represents the number of time steps of the time series.

In assessing the accuracy, the error between observed and forecasted quantities is estimated for arrival times and maximum tsunami amplitudes. However, one possible difficulty in establishing a standard metric is that each of these parameters has its own scale with significantly different ranges of values. For example, while a error in arrival time of a a few minutes can be considered reasonable (for instance, less than five), a variation in height of more than one meter can signify a large error. To provide a common comparison basis for all possible sensor configurations, each parameter error is adimensionalized by dividing by the reference provided by the observed dataset. In addition, it is possible that one parameter having a large error could bias the combined assessment to be implemented. Therefore, the error estimated is capped under the consideration that errors larger the observed value will be treated as equally significant. This is is implemented as follows

$$\Delta T_m = \min\left\{\left|\frac{T_m(\eta_{obs}) - T_m(\eta_{for})}{T_m(\eta_{obs})}\right|;1\right\}, \tag{6}$$

$$\Delta H = \min\left\{\left|\frac{H(\eta_{obs}) - H(\eta_{for})}{H(\eta_{obs}) - \eta(0)}\right|;1\right\}, \tag{7}$$

$$S_k = \min\{S_k;1\}. \tag{8}$$

where $m = 1,2$ applies for the different time travel parameters. In the case of the skill, $S_k = 1$ indicates that the magnitude of the error is comparable or greater than the observed values. Zero values mean a perfect fit on the indicator. In addition, whether each quantity is under or overestimated is not considered relevant and absolute values are used.

While it is possible to analyze each of these metrics independently, it is also desirable to identify the sensor configuration that yields the minimum combined error. However, it might be relevant for the user to give preference to one of the metrics above the others. Hence, to combine all estimators into a single quantitative estimate, a forecasting accuracy function is introduced

$$F_{i,j}(\Delta T, \Delta H, S_k) = \alpha \Delta T_{i,j} + \beta \Delta H_{i,j} + \gamma S_k i,j, \tag{9}$$

which allows quantifying the total error of the estimation at the forecasting point $j$ given an offshore array $i$. $\alpha, \beta$ and $\gamma$ are weights that allow for user-defined tuning of the relative importance of each parameter. The sum of the weights should add to one, as to preserve the comparison basis. As a first estimation, $\alpha = 0.40, \beta = 0.40$ and $\gamma = 0.20$ are considered, owing to the larger weight attributed to arrival time in previous research. Moreover, as to couple both definitions of arrival times, the error

estimator associated to this parameter was considered as the average of each percentage error, $\Delta T = 0.5(\Delta T_1 + \Delta T_2)$. In this way, it is possible to quantitatively compare the performance of all sensor arrays at any given forecast point. By introducing the saturation value, errors $\Delta T_m$ or $\Delta H$ exceeding 100% are not penalized in excess and do not bias the overall error, allowing highlighting the effect of the other parameters in the comparison.

However, it is of interest to test global accuracy of the method at several forecast points simultaneously. For instance, to evaluate how a given network configuration improves the predictive capability at several coastal sites. The overall performance is computed by simply adding up the individual results at forecast points of interest, given a sensor array $i$

$$EG_i = \frac{1}{N} \sum_{j=1}^{N} F_{i,j}(\Delta T, \Delta H, S_k), \tag{10}$$

10 where $N$ is the number of forecast points. Upon calculating the global error by means of Eq. 10 considering all possible offshore arrays, the candidate array is selected as that featuring the minimal global error $(\min(GE_i))$. As in the case of the Eq. 9, it is possible for Eq. 10 to be biased by a few forecast points. To this end, it is imposed that each forecasting point $j$ of the array should have an accuracy function value smaller than a certain threshold $(F_{i,j}(\Delta T, \Delta H, S_k) < \mu, \quad \forall j)$. This is equivalent to ensuring a minimal forecast capacity on each forecasting point $j$.

Regarding the construction of a sensor array, prior research indicates that two to four observation stations are capable of constraining the source parameters quite well if the stations are optimally located relative to the main tsunami energy beam, and adding more data does not significantly improve the inversion results (e.g. Percival et al., 2011; An, 2015). On the other hand, the relative distance of the sensors will depend on the time allowed to record the tsunami (e.g. Bernard et al., 2001).

20 Hence, given an earthquake, it is possible to define the area (henceforth termed listening area $A_l$) over which the tsunami waves have already propagated away from the source(e.g. Williamson and Newman, 2018). A minimum of two sensors must be considered inside this area. This equivalent to defining a data observation time, if the tsunami propagation speed is known. This time, termed $T_0$, represents the time the TWS allows for recording of the tsunami before performing an inversion.

25 To test the performance of the sensor arrays, a set of tsunami sources must be considered. While it is possible to test a large set of sources over a large domain as in Schindelé et al. (2008), here the focus is set on using sources located at the extreme of the area of interest, under the assumption that these will correspond to the worst scenario for the arrays owing to the decrease in sensitivity of detection as sensors are located further away of the main tsunami energy beam. Finally, once a sensor array has been defined, the candidate network is tested using other locations for synthetics sources and past events.

**3 Data and Sensor Array Design**

For the purpose of testing the possible sensor arrays, two tsunamigenic earthquakes, with seismic moments ($M_w$) 8.3 and 8.5 respectively, are considered as scenarios. These were determined by Cienfuegos et al. (2014) as representative events for the Northern Chile Gap. These are located just in front of Arica, near the Chile-Perú border, and just north of Peninsula Mejillones, as seen in Fig. 1. They also flank the rupture area of the 2014 Pisagua Earthquake. These scenarios were estimated from the interseismic coupling model of Chlieh et al. (2011), information on interseismic slip rates and convergence. The choice of the scenarios is based on the assumption that these constitute the worst case for performing an inversion. Hence, the optimal array should be capable of detecting each scenario and others located in between them.

A set of ∼ 1000 unit tsunami sources were distributed uniformly over an area spanning $160 \times 680$ km. Tsunamis were propagated using the tsunami model JAGURS (Baba et al., 2016) to eight coastal stations (coinciding with existing tide gages operated by the National Hydrographic and Oceanic Service of the Chilean Navy, SHOA, by its acronym in Spanish). A subset of these will constitute the forecast points where the accuracy functions $F_{i,j}$ are valuated. They are also propagated to an array of virtual observation points, each of which denote the possible location of a sensor. JAGURS is a numerical model for dispersive tsunami wave modeling. This is a parallel software which solves the nonlinear Boussinesq dispersive equations in spherical coordinates. To solve these equations, a leapfrog staggered-grid, finite-difference calculation scheme is used. Bathymetry considered GEBCO global data with spatial resolution of 30 arc seconds. All Green's functions were calculated in advance and stored in a database, from which appropriate functions were extracted and used for the inversion and forward calculation.

Although in operational conditions, the time series recorded at an actual sensor will also include other signals that could be considered as tsunami noise, these are not considered here under the premise that the approach used herein is to discriminate among sensor configurations, assuming the treatment of tsunami noise is equivalent across sensors. Similarly, it is noted that possible operational restrictions regarding sensor placement are obviated although they could be implemented straightforwardly.

[Figure]

**Figure 1.** (a-b). Sea surface deformation for the scenarios considered. Stars indicate epicenters. Black squares show virtual observation points offshore, red triangles indicate the forecasting points, and the yellow triangle represents the fixed sensor. (a) shows a $M_w$ 8.3 and (b) a $M_w$ 8.5 scenarios, located in the northern and southern end of the the Northern Chile Gap, respectively.(c-d) are the corresponding maps of tsunami energy. In (c-d), the final sensor configuration is shown.

The relative position of each sensor to the tsunami scenarios, determine the tsunami arrival time to the sensor but also the elapsed observation time of data usable for inversion. In existing inversion procedures, each sensor uses a different observation time, sufficient to gather at least a quarter of the initial tsunami waveform. Here, the observation time $T_0$ is defined as to determine an area such as any sensor located inside of this area could record at least half a tsunami wavelength ($0 < t < T_0$). This defines the listening area, $A_l$. It is noted that the observation time is also restricted by the tsunami arrival to the coast, in order to provide sufficient lead time for an eventual warning. As a starting point, $T_0$=10 min is used, consistent with the observed tsunami arrival for 2014 Pisagua tsunami (Catalán et al., 2015).

Each of the tsunami scenarios is propagated to determine its propagation time and estimate the listening area. Considering the time restriction imposed, there is relatively small region where the listening area $A_l$ of both scenarios overlap. Hence, it is possible to place a sensor that would serve both the northern and southern sections, located in the outer rise offshore of Iquique (see yellow triangle in Fig. 1). To find the location of the other sensors, the listening area for each scenario is discretized regularly every 0.25 arc degrees ($\sim 30$ km) in both latitude and longitude, which is significantly less than the tsunami wavelength in the area. Each node is considered a possible sensor location. In addition, the nodes are to be located at depths large enough for tsunami nonlinear effects to be considered negligible, thereby ensuring consistency with the assumption by the inversion algorithm. As a consequence, 99 and 113 possible nodes are defined in the northern and southern parts, respectively, denoted as black squares in Fig. 1. The difference in the number of nodes is due to the differences in the source dimensions, and in tsunami celerity arising from the bathymetry, which in unison determine different listening areas. A sensor array was defined by the pairing of the common sensor and each of the possible nodes.

To carry out the analysis, each scenario is propagated forward to all forecast points (FPs), using the Nonlinear Shallow Water Equations as implemented in the model COMCOT. The time series at nodes in deep water are considered as target (observed) time series to be used in the inversion process, whereas the use of coastal forecast points is aimed at establishing the quality of the assessment (see below). The use of a different tsunami model in propagating the signal and in preparing the database of Green's functions allows for differences in the target tsunami time series to those of the database, to reduce possible overfitting in the inversion. No other source of variability, such as noise, is considered. It is also assumed that tsunami-tide interactions do not play a significant role in this area.

For each of the 212 sensor arrays (99 and 113, for each scenario), these observed tsunami time series are used to invert the tsunami source. Once a source is determined, the linear combinations of the Green's functions consistent with the source weights are used to estimate the forecast time series in the coastal points. It is of note that the use of linear superposition at shallow coastal points might be subject to inaccuracies arising from neglecting nonlinear interactions, and other process such as bathymetry–induced effects such as resonance. However, it is expected that omitting these process will affect the accuracy but not does limit discriminating among sensor configurations. Moreover, coastal points are located at 200 m. depths to reduce the effect of nonlinearity.

[Figure]

**Figure 2.** Spatial distribution of the individual error estimates, as a function of sensor arrays in the northern section. (a) $\Delta T$ (Eq. 6); (b) $\Delta H$ (Eq. 7); (c) $S_k$ (Eq. 8); and (d) Total forecasting accuracy function, $F_{i,j}$ (Eq. 9) estimated using as reference the coastal forecast point in Arica. Star indicates the scenario epicenter, the thin black line is the location of the trench, the yellow triangle denotes the location of the fixed sensor, and each grid point in the color map corresponds to a movable sensor.

Consequently, the observed (from the full tsunami simulations using COMCOT) and the forecasted (from the linear combination using inverted source solutions) are compared to assess the accuracy of the solution. To this end, individual and aggregate error estimates (Eqs. 6-10) are calculated .

**4   Results**

[revised manuscript text omitted]

In addition, it is possible that for actual implementations, the selection of a sensor may provide additional restrictions. For example, the ability to transmit data in real time constraining to line-of-sight placement, or communications coverage; or deployment away from the trench to reduce the effect of seismic noise and coseismic signals, among many possible restrictions.
10 Such restrictions were not considered herein as they can be sensor-specific, though they could be easily incorporated in the method by simply restricting locations where a sensor can be deployed.

The methodology proposed considered two characteristic scenarios located at the extremes of the area of interest. It is relevant to evaluate the capabilities of the proposed network in other cases to ensure that the proposed configurations do offer good
15 performance not only for the target scenarios. To this end the southern event ($M_w$ 8.5) was modeled in different epicentral locations along the subduction zone, every 0.5 arc degree along strike (see Fig 5). In addition, the listening time was changed ($T_0$=10, 15 and 20 min) to investigate the effect of record length on the forecast.

Fig. 6 shows the matrices of $F_{i,j}$ at each forecast point $j$ as a function of the epicenter location, considering the final array
20 configuration proposed, for different listening times, $T_0$. Regarding the latter, the aggregate error decreases as $T_0$ is increased, as expected, where aggregate errors can reach values as low as 15% when $T_0 = 20$ min (compare Fig. 6a and c). In the case

[Figure]

**Figure 6.** Forecast accuracy function $F_{i,j}$ as a function of reference forecast point $j$ (vertical axis), considering different target scenario locations (horizontal axis) and listening time, $T_0$. (a) 10 min, (b) 15 min and (c) 20 min.

of Chile, where the seismic zone is located very close to the coast, this may prevent the use of this information as a trigger for early warning, but it could be considered in the later stages of the emergency cycle. For instance, this information can be used to refine initial hazard assessments derived by other means, such as the existing database of precomputed scenarios.

5  Regarding the sensor array performance for different scenarios, it is possible to observe a decrease in performance as the scenario is closer to the reference forecast point. This can be seen as a poor accuracy in the northern FPs (Arica to Iquique) for events with epicenters at latitudes -19.5° and -20° (see dark blue data in the upper left corner of Fig. 6a), and a worsening of the accuracy as the epicenters are located further south (see the evolution of the error for Tocopilla). This is due to the forecast points being placed close or inside the zone of the predicted coseismic deformation for the actual event, which prompts an

10 early arrival in the observed signal. However, owing to the short listening time, the tsunami source solution results in a source of small spatial dimensions (see for example Fig. 7b, although for a different scenario). As result, the arrival time yields a large error. In the other cases, $F_{i,j}$ averages 40% which is considered a good performance for this observation time. As the observation time is increased, the error in the prediction drops significantly and values as low as $F_{i,j} \approx 20\%$ error can be obtained.

15  As an additional test for evaluating the potential performance of the sensor array, the tsunami of Pisagua on April 1st, 2014 ($M_W 8.1$) was used as source scenario. It is noted that this event has a smaller magnitude than the scenarios used in designing the network, and also that it has a non-uniform slip distribution, with two patches of slip. Synthetic tsunami waveforms at the location of the sensor array were obtained by a numerical simulation of the tsunami using the initial rupture model proposed by Hayes et al. (2014). As before, the accuracy of the assessment is evaluated using the coastal forecast points and different

20 listening times, although in this case, the actual tide-gage records are considered. These were obtained from the IOC Sea Level

Monitoring Facility website.

Results are summarized in Table 2 and the initial surface solutions are shown in Fig. 7. It can be seen that within 10 min, the performance of the method is inadequate, with large errors and providing an inverted initial surface that only detects a localized source. As the observation time is increased, the errors decrease significantly, with the reconstructed sea surface condition now having an appropriate extension but a smaller peak displacement. The solution is improved for $T_0 = 20$ min, with errors as low as 8% (40% on average), and the solution includes traces of the secondary peak on the north west of the rupture. However, as mentioned before, such a sensor array would have only provided a timely assessment for locations outside the main rupture zone, since the observed tsunami arrival at the tide gages in Iquique and Pisagua was less than 15 min (An et al., 2014; Catalán et al., 2015).

**Table 2.** Summary of tsunami forecasting accuracy considering different data observation time for 2014 Pisagua event.

| Forecasting Point | Observation time | | |
|---|---|---|---|
| | 10 [min] | 15 [min] | 20 [min] |
| Arica | 0.66 | 1.00 | 1.00 |
| Pisagua | 0.83 | 0.90 | 0.70 |
| Iquique | 0.94 | 0.76 | 0.57 |
| Patache | 1.00 | 0.96 | 0.94 |
| Tocopilla | 0.83 | 0.77 | 0.79 |
| Mejillones | 0.59 | 0.55 | 0.54 |
| Antofagasta | 0.57 | 0.57 | 0.57 |

This suggests that the proposed network is capable of identifying smaller events with non-uniform slip reasonably well, in reasonable time. It is of note also that the location of this scenario does not coincide with either of the scenarios used to design the network. However, two of the sensors are located close but not directly into the main energy beam. When considered in unison, these two tests suggest that the sensor network considered would be appropriate for tsunamis generated by earthquakes of magnitude similar and larger than that of the initial target scenarios. It is possible, however, that a different network would be obtained if different target scenarios were used for the design. Nevertheless, the methodology allows for identifying a suitable sensor array. This stresses that one relevant step prior to the implementation of the methodology would be to determine these target scenarios by other means. In the present case, the choice was based on data available for the Northern Chile Gap.

On the other hand, the methodology expands on previous research by incorporating quantitatively different tsunami parameters in a cost function which can be minimized. However, the selection of the weights ($\alpha, \beta$ and $\gamma$ ) in the cost function $F_{i,j}$, Eq. 9, was arbitrary. To test the influence of these weights in the assessment, a sensitivity analysis was carried out. Each weight

[Figure]

**Figure 7.** Initial sea surface displacement obtained from different listening times for the 2014 Pisagua tsunami. (a) Target result resulting from using the slip distribution of Hayes et al. (2014) as source scenario. (b) – (d) Distribution of the initial sea-surface displacement estimated by tsunami waveform inversion considering $T_0 = 10$, 15 and 20 min. (e) – (f) Comparison of observed (red lines) and calculated (blue lines) waveforms in the three sensors. Vertical green lines in (e), (f) and (g) indicate the listening time.

was modified by up to $\pm 0.1$, with a 0.05 step. In addition, the case where only one parameter was used is also considered, yielding 23 weight combinations as shown in Table 3.

As a first step, it is analyzed whether modifying the weights induces a change in the selection of the sensor array. In Fig 8a-d, sample spatial maps of the global error $EG$ are shown for some of the weight combinations (baseline combination, and

5    combinations 10, 12 and 20, see Table 3 for details). It can be seen that although there is a variation of the value of the error, and also some variation of the spatial distribution of the magnitudes, the overall structure is consistent. The notable exception is combination 20, which only considers $\alpha = 1.0$, thus considering only the error in arrival times. Fig. 8e shows the value of the global error $EG$ for each array configuration, as a function of the weight combination. For all combinations, the minimum error is obtained for the array 62, which is the one initially determined. This could be due to several factors. For instance, that

10    the sensitivity range used in this test did not suffice to alter the result. However, even when only one parameter is considered,

[Figure]

**Figure 8.** (a-d) Sample spatial maps of the global error, $EG$, as a function of the combination of weights. From left to right, combinations 1, 10, 12 and 20, respectively. (e) Matrix of the error $EG$ for each sensor array as a function of the combination of weights. Array configurations are numbered in accordance to the grid location, from west to east, and then from north to south.

the solution remains unaltered. Consequently, it could be possible that the solution is controlled by only one parameter. For instance, the error in amplitude $\Delta H$ yields the minimum errors in several combinations (see red colors for combination 21 in Fig. 8e). However, there are some instances where it is the arrival time the parameter yielding the minimum error (see combination 20). Therefore, it is concluded that the use of a cost function combining both parameters maximizes the ability to capture well the overall structure of the tsunami signal. Moreover, the choice of weights proposed appears to be a good compromise among them.

Regarding the effect of the weighting on the accuracy assessment, it was also tested how much do the accuracy estimates for the Pisagua tsunami change with the different combinations of weights. The results are shown in Fig. 9, where it can be seen that in general, most forecast points have a relatively stable assessment, typically varying by less than 10% in the error estimate. However, the situation changes significantly if only one parameter is used. For instance, if only time is con-

[Figure]

**Figure 9.** Forecasting accuracy by means of Equation 9 considering different weight combinations showed in the table 3 for the tsunami of Pisagua 2014.

sidered (combination 20), most forecast points worsen their assessment, as in Arica and Iquique. On the other hand, Patache and Antofagasta (which are overlapped since they have a similar, but not equal error, close to the saturation limit), shows a large drop in the error associated with the arrival time. The situation reverses when only the skill is considered (combination 22), when Arica and Iquique improve the estimation but Patache worsens significantly. Therefore, it is found that the use of
5  multiple parameters is essential in producing a more complete estimation of the error for determining the optimal configuration.

The above results suggest the methodology is capable to deliver a working tsunami observation system comprised by just three sensors. Although the general rule that the spacing between sensor should be in the range $\sim$ 200-400 km (Bernard et al., 2001) is somewhat preserved, the spacing between sensors differs, between 110 and 225 km, approximately. This yields a
10  less dense network as the one proposed by either Schindelé et al. (2008) and Omira et al. (2009). Moreover, the methodology

**Table 3.** Parameter space of weighting functions used.

[revised manuscript text omitted]

Saunders and Haase: Augmenting onshore GNSS displacements with offshore observations to improve slip characterization for Cascadia subduction zone earthquakes, Geophysical Research Letters, 0, https://doi.org/10.1029/2018GL078233, https://agupubs.onlinelibrary.wiley.com/doi/abs/10.1029/2018GL078233, 2018.

Schindelé, F., Loevenbruck, A., and Hébert, H.: Strategy to design the sea-level monitoring networks for small tsunamigenic oceanic basins: the Western Mediterranean case, Natural Hazards and Earth System Sciences, 8, 1019–1027, https://doi.org/10.5194/nhess-8-1019-2008, 2008.

Soulé, B.: Post-crisis analysis of an ineffective tsunami alert: the 2010 earthquake in Maule, Chile, Disasters, 38, 375–397, https://doi.org/10.1111/disa.12045, 2014.

Spillane, M. C., Gica, E., Titov, V. V., and Mofjeld, H. O.: Tsunameter network design for the U.S. DART arrays in the Pacific and Atlantic Oceans, NOAA Technical Memorandum OAR PMEL-143, National Oceanic and Atmospheric Administration, 2008.

Titov, V.: Tsunami forecasting, in: , Chapter 12 in The Sea, Volume 15: Tsunamis, Harvard University Press, Cambridge, MA and London, England, 371–400., vol. 212, pp. 629 – 636, https://doi.org/10.1016/j.proeng.2018.01.081, 7th International Conference on Building Resilience: Using scientific knowledge to inform policy and practice in disaster risk reduction, 2009.

Tsushima, H., Hino, R., Fujimoto, H., Tanioka, Y., and Imamura, F.: Near-field tsunami forecasting from cabled ocean bottom pressure data, Journal of Geophysical Research, 114, B06 309, https://doi.org/10.1029/2008JB005988, 2009.

Tsushima, H., Hino, R., Tanioka, Y., Imamura, F., and Fujimoto, H.: Tsunami waveform inversion incorporating permanent seafloor deformation and its application to tsunami forecasting, Journal of Geophysical Research, 117, B03 311, https://doi.org/10.1029/2011JB008877, 2012.

Williamson, A. L. and Newman, A. V.: Suitability of Open-Ocean Instrumentation for Use in Near-Field Tsunami Early Warning Along Seismically Active Subduction Zones, Pure and Applied Geophysics, https://doi.org/10.1007/s00024-018-1898-6, 2018.